# Building, Reusing, and Generalizing Abstract Representations from Concrete Sequences

**Shuchen Wu**
Helmholtz Munich
Max Planck Institute for Biological Cybernetics
shuchen.wu@tuebingen.mpg.de

**Mirko Thalmann**
Institute for Human-Centered AI
Helmholtz Munich
mirko.thalmann@helmholtz-munich.de

**Peter Dayan**
Department of Computational Neuroscience
Max Planck Institute for Biological Cybernetics
dayan@tuebingen.mpg.de

**Zeynep Akata**
Helmholtz Munich
Technical University of Munich
zeynep.akata@helmholtz-munich.de

**Eric Schulz**
Institute for Human-Centered AI
Helmholtz Munich
eric.schulz@helmholtz-munich.de

## Abstract

Humans excel at learning abstract patterns across different sequences, filtering out irrelevant details, and transferring these generalized concepts to new sequences. In contrast, many sequence learning models lack the ability to abstract, which leads to memory inefficiency and poor transfer. We introduce a non-parametric hierarchical variable learning model (HVM) that learns chunks from sequences and abstracts contextually similar chunks as variables. HVM efficiently organizes memory while uncovering abstractions, leading to compact sequence representations. When learning on language datasets such as babyLM, HVM learns a more efficient dictionary than standard compression algorithms such as Lempel-Ziv. In a sequence recall task requiring the acquisition and transfer of variables embedded in sequences, we demonstrate HVM's sequence likelihood correlates with human recall times. In contrast, large language models (LLMs) struggle to transfer abstract variables as effectively as humans. From HVM's adjustable layer of abstraction, we demonstrate that the model realizes a precise trade-off between compression and generalization. Our work offers a cognitive model that captures the learning and transfer of abstract representations in human cognition and differentiates itself from LLMs.

## 1 Introduction

Abstraction plays a key role in intelligence (Konidaris, 2019). Philosophers traditionally view abstract ideas as formed by identifying commonalities across experiences, distilled from concrete impressions grounded in perception (Kant, 1998; Fichte, 2005; STERN, 1977). Psychologists suggest that abstraction arises from personal experiences, such as forming the concept of "whiteness" by observing various white objects (Yee, 2019; Barsalou, 1999). Abstract concepts are thought to build on concrete concepts and on top of the previously learned abstractions, thereby varying in complexity (Cuccio & Gallese, 2018; Van Oers, 2001; Collins & Quillian, 1969; Piaget, 1964). The ability to abstract, which is often seen as a human-specific trait, enables reasoning, generalization, and problem-solving in novel contexts (Ohlsson & Lehtinen, 1997; Dehaene et al., 2022; Duncker, 1945).

We hypothesize that the world contains patterns across scales of time and abstraction. Intelligent agents-facing sequences with nested hierarchical structures- need to model these structures to store, process, and interact in such environments. As a rational strategy, intelligent agents shall characterize the temporal structure via chunking and characterize the abstract structure via identifying different

items that play similar roles. To explore these operations, we design a generative model that produces sequences nested with hierarchical structure and propose an approximate recognition model that conjunctively learns to chunk and abstract.

We go beyond previous proposals that introduce chunking as a mechanism for learning to compose complex structures from elementary perceptual units (Gobet et al., 2001; Miller, 1956; Wu et al., 2022) and propose a cognitive model that combines chunking and abstraction in one single system. The model uses abstraction in two key ways: first, by identifying shared features to facilitate efficient pattern retrieval; and second, by categorizing different sequential items that appear in the same context, much like a variable in a computer program. These paired mechanisms enable the model to parse sequences into chunks and form abstract patterns based on both concrete and previously learned abstractions, allowing for compact representations while revealing recurring patterns at the level of abstract categories. Consequentially, increasingly complex abstract patterns can be discovered layer by layer.

We first demonstrate the benefits of abstraction in memory efficiency and sequence parsing by comparing our algorithm with previous chunking models and other dictionary-based compression methods. Then, we show that the model exhibits human-like signatures of abstraction in a memory experiment requiring the transfer of abstract concepts. In the same experiment, we contrast the model's generalization behavior with large language models (LLMs). Additionally, we demonstrate the connection between abstraction level and abstract concept transfer by varying the level of abstraction as a parameter in the model. Our work offers a cognitive model that captures the learning and transfer of abstract representations in human cognition and differentiates itself from the behavior of artificial agents.

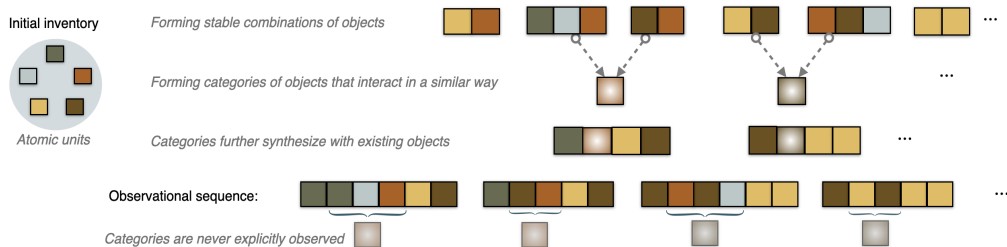

Figure 1: Generative Model. The inventory is initialized with a set of atomic units, which randomly combine into objects. Some objects are randomly selected to become categories. Categories of existing objects have similar interaction properties. The dashed arrow points from those objects to their corresponding categories. These categories further combine with existing objects. An observational sequence is composed of objects randomly sampled from the existing inventory. Categories are latent and never explicitly observed but are manifested in one of the denoting objects.

## 2 GENERATING SEQUENCES WITH OBJECTS THAT CONTAIN HIERARCHICAL ABSTRACT STRUCTURES

Hierarchical structures have been long presumed to characterize natural sequences — such as molecules composed of chemical elements or the hierarchical nature of language (Abler, 1989; Sportiche et al., 2013; von Humboldt & Losonsky, 1999). To study models respect hierarchical structure, we take a standard approach starting from a conformative generative model (Teh, 2006; Milan et al.). We design a probabilistic generative model that generates sequences by sampling objects from an inventory which contains a hierarchical structure.

The generative model creates an inventory of recurring objects over $d$ iterations of expansion. As illustrated in Figure 1, the inventory starts with a set of atomic units $\mathbb{A}$. On each iteration, a novel object or category is created equiprobably. A new category (graded node) is created by pointing to a random selection of objects from the inventory up to the moment (objects are treated disjunctively). A new object is created by concatenating a random selection of pre-existing objects or categories from the existing inventory. After the inventory has been expanded up to the $d$-th iteration, the independent occurrence probability in the sequence is sampled from a flat Dirichlet distribution

$f(p(c_1), ..., p(c_{|\mathbb{A}|}); \alpha_1, ..., \alpha_{|\mathbb{A}|}) = \frac{1}{\mathbf{B(a)}} \prod_{i=1}^{|\mathbb{A}|} P(c_i)^{\alpha_i - 1}$, $\alpha_i = 1 \forall i$ and assigned to each object in the inventory. Similarly, a probability is sampled from a flat Dirichlet to assign the occurrence probability of the objects within each category. More details can be found in the Appendix A.2.

To create an observational sequence, objects are randomly drawn from the inventory one after another with the assigned probability until they reach the desired sequence length. If the sampled object contains an embedded category, one of the objects corresponding to the category is sampled. This process is repeated recursively until all categories are replaced by specific objects, resulting in a sequence of discrete atomic units.

## 3 LEARNING ABSTRACTIONS FROM SEQUENCES

We ask what computational principles could help an agent discover objects and categories from such observational sequences without supervision. We propose that two mechanisms suggested by the cognitive literature are vital: chunk proposal and variable discovery. Chunking concatenates learned objects and forms new ones; variable discovery groups chunks with similar interaction properties into a category. We build on top of the hierarchical chunking model (HCM) (Wu et al., 2022), which learns a belief set $\mathbb{B}$ of a chunk dictionary $\mathbb{C}$ from discrete sequences. We expand the model so that it can also learn variables while improving the memory efficiency of the model. By identifying stably recurring entities as chunks and grouping similar entities into categories as variables, the agent can learn a structured inventory of identifiable patterns and use these patterns as entities to parse the sequence, leading to a more compressed factorization of perceptual sequences.

HVM learns a belief set $\mathbb{B}$ that contains both a dictionary of chunks $\mathbb{C}$ and variables $\mathbb{V}$. The variables are proposed as abstract entities based on the transition and marginal counts. As shown in Figure 2a, each variable $v \in \mathbb{V}$ denotes a set of chunks $E(v) = \{c_j\}$, $c_j \in \mathbb{C}$. The model also learns the probability of each chunk that a variable denotes $\forall v \in \mathbb{V}$, $\sum_j P(v \rightarrow c_j) = 1$, $c_j \in \mathbb{C}$.

**Parsing the sequence one chunk at a time**   A fundamental feature of the model is to parse sequences in chunks as the basic cognitive units (Miller, 1956). Along with each parse $t$, the transition counts between the previous $c_L$ and next chunks $c_R$ are recursively updated: $T_{ij}(t + 1) = T_{ij}(t) + [i = c_L][j = c_R]$ ($[\cdot] = 1$ if the argument is true and $0$ otherwise) and along with the identification frequency of each parsed chunk $M_i(t + 1) = M_i(t) + [i = c]$. When modeling human behavior, each entry of $M$ and $T$ multiplies with a memory decay parameter $\theta$ per parsing step. The probability of observing a sequence of parsed chunks $c_1, c_2, ..., c_N$ then becomes $P(c_1, c_2, ..., c_N) = \prod_{c_i \in \mathbb{C}} P_{\mathbb{C}}(c_i)$.

To parse a sequence, HCM iteratively chooses the biggest chunk amongst its learned dictionary $\mathbb{C}$ consistent with the upcoming sequence. The end of a previous parse initiates the next parse. As the dictionary size $|\mathbb{C}|$ increases, searching for the biggest consistent chunk becomes computationally expensive (Schreiber et al., 2023). In HVM, we introduce one notion of abstraction as finding *commonalities amongst memory items* to organize memory structure and speed up chunk retrieval during parsing. Memory items in the learned dictionary are organized into a hierarchical parsing graph that connects chunks with their common prefixes. Hence, all children chunks are different except for sharing a common prefix from the parent. The parsing graph arranges the chunks in $\mathbb{C}$ into a prefix Trie structure (Figure 2b), reflecting the cue-based, content-addressable nature of memory retrieval (Cunnings & Sturt, 2018; Dotlačil, 2021; Anderson, 1974). This design reduces search time to retrieve a chunk as arranged in the parsing graph, commonly used in predictive text or auto-complete dictionaries to speed up search steps (Fredkin, 1960). At every parsing step, HVM identifies the deepest chunk in the parsing graph that is consistent with the upcoming sequence. The end of the previous parse initiates the next parse. The search process would take the time complexity of $O(D)$, scaling with the depth $D$ of the tree compared to HCM time complexity of $O(|\mathbb{C}|)$ scaling with the size of $\mathbb{C}$. The appendix shows a guarantee to reduce the number of parsing steps for chunk retrieval A.4.

**Learning chunks**   From parsed sequences, the model estimates the occurrence probability of chunk via the entries of $M$ as $P_{\mathbb{C}}(c_i) = \frac{M_i}{\sum_{c_j \in \mathbb{C}} M_j}$. The model starts with the null hypothesis $\mathcal{H}_0$ that all consecutively parsed chunks $c_L$ and $c_R$ are statistically independent $P(c_L, c_R) = P(c_L)P(c_R)$. If a significant correlation is found between consecutively parsed chunk pairs (with $p = 0.05$), the null

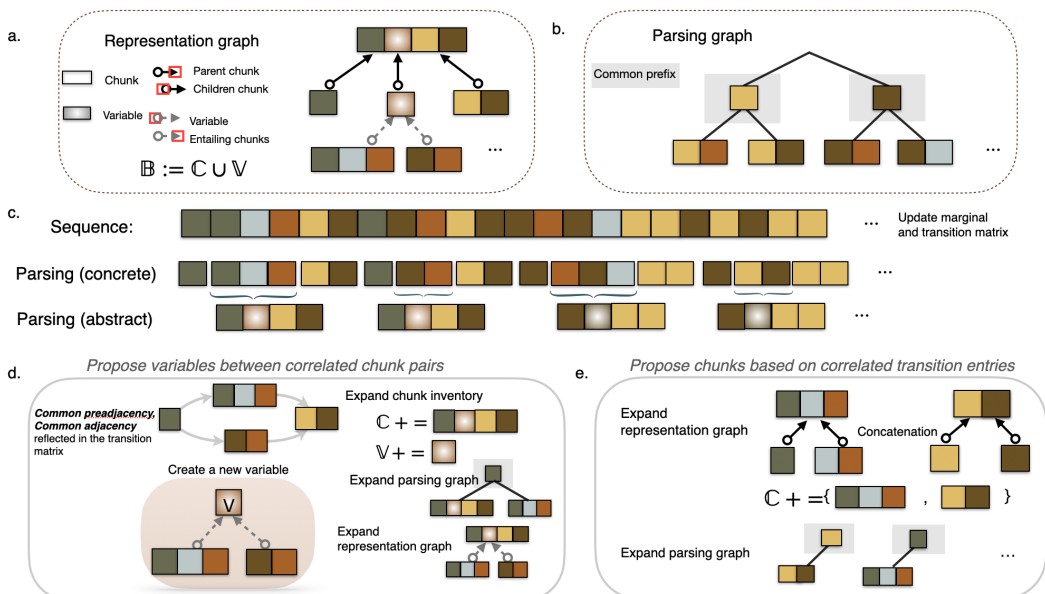

Figure 2: HVM builds up a representation graph and a parsing graph. a. A representation graph contains the learned chunks (nodes with solid colors) $\mathbb{C}$ and the contained variables (nodes with gradient colors) $\mathbb{V}$. Black arrows denote concatenation. Gray and dashed arrows point from chunks of the same category to the variable node that denotes this category. b. Abstraction organizes memory in a parsing graph. Nodes shaded by gray are abstract chunks being the common prefix intersection of its children. During parsing, to allocate to the matching chunk in $\mathbb{C}$, the model starts searching from the root of the parsing tree and traverses to the deepest node consistent with the upcoming sequence content. c. Upon completing each sequence parse, HVM updates counts on chunk frequencies and transitions and proposes inventory expansions, enabling a layer-wise discovery of recurring patterns per iteration, from specific to the more abstract. d. HVM proposes variables amongst correlated consecutive chunk pairs. e. Chunk proposal.

hypothesis is rejected, and the pair is concatenated into a new chunk, $c_L \oplus c_R$, which is then added to the dictionary $\mathbb{C}$ as a new entry. Upon creating each new chunk, an edge from the left parent chunk (sharing the same prefix) connects to the newly created chunk in the parsing graph. When no parent chunk can be found, an ancestor node is created in the parsing graph, usually the sequence's atomic units. We prove in the SI section A.3 that under restricted conditions in which the sub-objects constituting the object in the ground truth are parsed faithfully by the model as sub-chunks, then these sub-chunks will be eventually proposed to concatenate into one chunk.

**Learning variables**  A variable denotes distinct observations appearing in the same context (here defined as distinct chunks sharing preceding and succeeding chunks). Given consistent parsing of subchunks as object combination components in the generative model, the true chunks that belong to the same category will necessarily appear in the preadjacency and postadjacency entries (more explanation in SI A.3). HVM proposes variables amongst the significantly correlated chunks ($p \leq 0.05$) in the transition matrix consistent with this structure. For example, consider the case that the sequence includes A-ABC, A-DC, and ABC-ED, DC-ED. This suggests that ABC and DC play similar roles relative to A (coming after) and ED (before). Therefore, ABC and ED can be proposed as a new variable V to represent the abstract property that captures the distinct entities. V denotes ABC and DC, and the model identifies V if either ABC or DC is identified during parsing, as illustrated in Figure 2d and Figure 7 in SI.

Among the correlated consecutive chunk pairs, the model identifies all eligible transitions by intersecting the post-adjacency columns with the pre-adjacency rows, proposing a set of chunks $E(v) = c_1, c_2, ..., c_j$ represented by a new variable $v$. This variable proposal is accepted if the set of chunks that satisfy the condition $T_{min} \leq |E(v)| \leq T_{max}$ and $\sum_{c_i} M(c_i) \geq freq_T$. A variable

denotes a set of chunks $E(v)$ and is identified if any of the chunks that it denotes is parsed (Figure 2c). Together, the common preceding chunk, in conjunction with the newly proposed variable, followed by the common succeeding chunk, is concatenated and proposed as a novel chunk $c_L \oplus v \oplus c_R$ to add to the inventory $\mathbb{C}$. An edge that connects $c_L$ to the new chunk $c_L \oplus v \oplus c_R$ to be added to the parsing graph, as illustrated in Figure 2d. Two variables are merged into one if they share the same preceding and succeeding chunks. During parsing, a chunk is consistent with the sequence if any of its included variables contain a denoting chunk that is consistent with the sequence. Upon identifying a chunk during parsing, the marginal and transition count for both the chunk and the immediate variables that the chunk belongs to is incremented.

**Representation cleaning**  Upon completion of each sequence parsing, the model uses correlations to propose new variables and chunks. It then uses the expanded inventory for the next parse. Unused variables and chunks are deleted upon the completion of each parsing iteration.

# 4  RESULTS

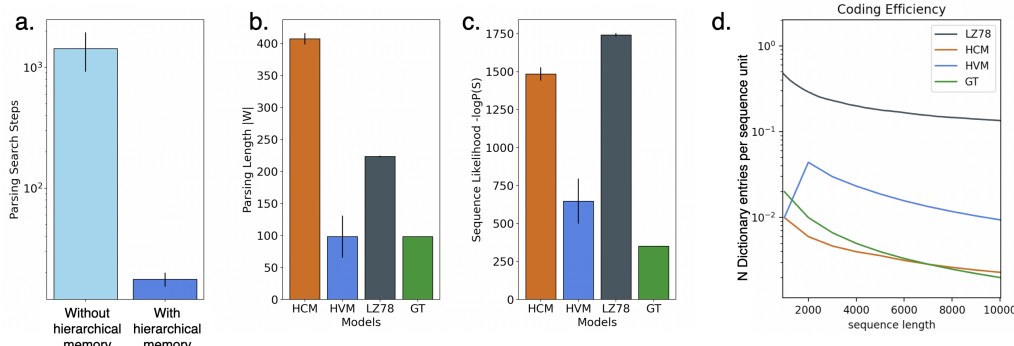

Figure 3: a. The effect of hierarchical memory structure on parsing search steps. b. Comparison between models in terms of sequence length after parsing. GT denotes the ground truth sequence length by the generative model. c. Model comparison based on sequence likelihood. d. Model comparison based on coding efficiency. Example model comparison with sequence length $|S| = 1000$, nested hierarchy depth $d = 30$, atomic set of size $|A| = 10$.

## 4.1  MODEL EVALUATION

HVM is an approximate inverse of the generative model, as in practice, learning the ground truth hierarchical patterns that generate observed data is a nonidentifiable problem (Post, 1946; Greibach, 1968). Therefore, we use a set of measures for model evaluation, focusing on parsing search steps, sequence length, sequence negative log-likelihood, and encoding efficiency. We trained models on sequences generated by the hierarchical generative model until convergence. For each iteration, the models parse the entire sequence using the dictionary updated from the previous iteration and propose novel chunks/abstractions upon completing the parse. While HVM learns both chunks and variables per iteration, HCM learns only chunks and no variables. Previous research has related human memory in specific tasks to lossy compression (Nassar et al., 2018), with abstraction playing a crucial role in successful learning and transfer in sequence memory recall tasks. For reference, we also compared the models with LZ78, an off-the-shelf compression algorithm underlying compression schemes such as GIF and PNG, which also parses sequences into chunks and builds a dictionary to compress sequence data (Ziv & Lempel, 1978). Comparisons are shown in Figure 2.

**Organizing chunks in a parsing graph based on common prefix reduces parsing search steps**
We compared HVM with HCM, which does not organize memory into a hierarchical parsing graph. Figure 3 a. shows that organizing chunks in a parsing graph dramatically reduces the average number of parsing search steps, i.e., the number of search steps needed for the model to locate the biggest chunk per parse. This search step now scales with the depth of the prefix tree compared to the size of the inventory in HCM. More discussions can be found in A.4.

**Discovering bigger recurring patterns** We compared HVM with its counterpart, HCM, which does not learn variables at the end of the learning iteration. Figure 3 b compares sequence length $|W|$ after parsing. HVM transforms the sequence $|S|$ into a code with a much smaller size $|W|$ compared to HCM and LZ78 and is on par with the ground truth (GT). Learning variables that denote distinct chunks occurring in similar contexts helps HVM to learn bigger patterns underlying the sequence on the description level of these variables.

**Parsing sequences with higher likelihood** Figure 3 c compares the negative log likelihood $(-\log P(S) = \prod_{c_i \in \mathbb{C}} P(c_i))$ of the parsed sequence upon convergence. HVM parses sequences more efficiently, as variables recur more frequently in sequences than their representing chunks, and therefore, the model that learns variables parses the sequence with higher likelihood.

**Encoding more efficiently** In many applications, the size of the dictionary affects compression efficiency (Navarro & Mäkinen, 2007; Ferragina & Manzini, 2005). We compare the encoding efficiency, i.e., the ratio between the dictionary size (number of entries) and the original sequence length $S$. Figure 3d shows the relation between the ratio and the sequence length. To encode a sequence of the same length, LZ78 creates a bigger dictionary than HVM and HCM. Models that harness the embedded hierarchical structure in sequences encode sequences more efficiently and learn a smaller dictionary to encode the same sequence length.

## 4.2 LEARNING FROM REAL-WORLD SEQUENTIAL DATA

| Data Domain | Model | Compression Ratio | NLL | Coding Efficiency |
|---|---|---|---|---|
| CHILDES (MacWhinney, 2000) | LZ78 | 0.38 | 2837.50 | 0.34 |
| | HCM | 0.51 | 2783.71 | 0.05 |
| | HVM | 0.36 | 1953.01 | 0.06 |
| BNC (BNC Consortium, 2007) | LZ78 | 0.39 | 3136.67 | 0.37 |
| | HCM | 0.65 | 3591.60 | 0.06 |
| | HVM | 0.50 | 3108.33 | 0.08 |
| Gutenberg (Gerlach & Font-Clos, 2020) | LZ78 | 0.39 | 3156.61 | 0.37 |
| | HCM | 0.69 | 3770.36 | 0.06 |
| | HVM | 0.54 | 3252.84 | 0.12 |
| Open Subtitles (Lison & Tiedemann, 2016) | LZ78 | 0.41 | 3395.09 | 0.39 |
| | HCM | 0.74 | 4151.89 | 0.07 |
| | HVM | 0.63 | 3764.48 | 0.07 |

Table 1: Model comparison on alternative sequences. NLL stands for negative log-likelihood.

Going beyond artificially generated sequences, we evaluate HVM, HCM, and LZ78 in text domains from the BabyLM language dataset, which contain text snippets from a collection of data domains such as conversational child-directed speech, narrative stories, and other linguistically rich environments Warstadt et al. (2023). For each data domain, we took random snippets of 1000 characters and calculated evaluation metrics, including compression ratio (the number of tokens before compression divided by the number of tokens after compression), sequence complexity (negative log-likelihood), and compression efficiency (the length of the compressed sequence divided by the number of dictionary entries).

As shown in table 1, LZ78 performs well in terms of compression efficiency, which is expected given its design purpose. However, the HVM model exhibits notable advantages when considering alternative evaluation metrics. Specifically, for text across the four data domains analyzed, cognitive models that incorporate the hierarchical structure of data outperform traditional compression methods in terms of encoding efficiency. This is because LZ78, which does not make strong assumptions about sequence structure, tends to create redundant entries in its dictionary. While this redundancy reduces the sequence length after compression, it introduces many infrequently used entries. In some data domains, HVM outperforms LZ78 in compression ratio and negative log-likelihood as well. Amongst all domains, HVM compresses the sequence further and parses the sequence with lower complexity than HCM.

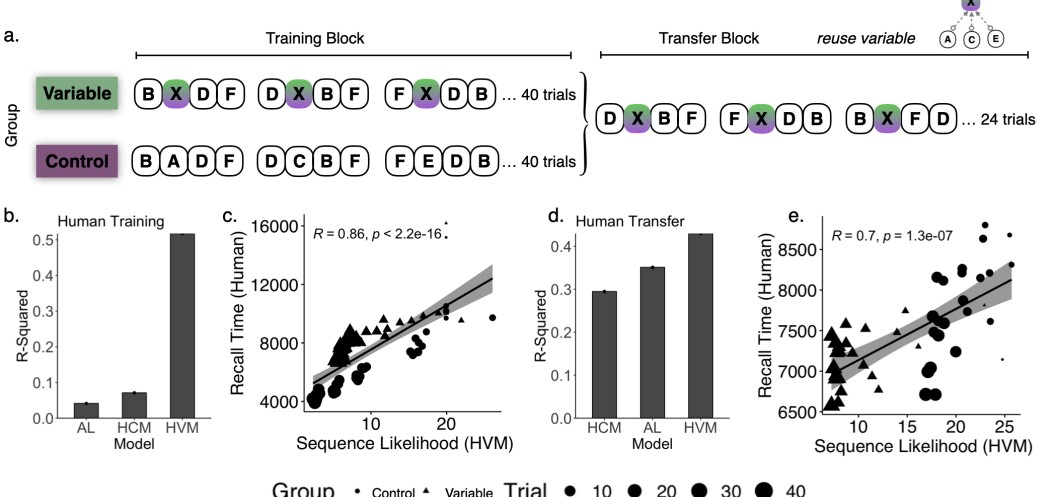

Figure 4: a. A sequence memory task that demands transferring a variable x from the training to the test block. b., c., d., e., Complexity (negative log-likelihood) progression of HVM correlates with human memory recall time in the training (b., c.) and the transfer block (d., e.). Memory decay parameter $\theta = 0.996$. Marker size increases with trial number.

### 4.3 HVM RESEMBLES HUMAN SEQUENCE MEMORIZATION AND TRANSFER

We take the relation between transfer to learning abstractions to the setting of a memory experiment that demands learning and transferring variables (Wu et al., 2023). We ask whether HVM resembles aspects of human learning while enjoying the advantage of learning an interpretable dictionary.

In the experiment, 112 participants were instructed to recall a sequence of presented colors from memory (Figure 4 a). Participants were randomly assigned to a variable group and a control group. During the training block (40 trials), the control group was instructed to remember a fixed sequence BADF DCBF FEDB for each trial (each letter denotes a distinct color). In contrast, the variable group was instructed to remember sequences overlapping in 9 of the 12 colors: BXDF DXBF FXDB. A variable X, however, appeared at serial positions 2, 6, and 10, which could be occupied with the letters A, C, or E with equal probability for each trial. After the training block, both groups were then tested on the transfer block. The transfer sequence overlapped with the training sequence only at the variable positions: DXBF FXDB BXFD. The time participants took to recall the entire sequence for each trial was recorded. Previous studies suggest human recall time relates logarithmically to the perceptual predictability of the sequence (Carpenter & Williams, 1995; Anderson & Milson, 1989; Smith & Levy, 2013; Elman, 1990). We compare human recall time to the model's negative log-probability (likelihood) evaluated on the instruction sequence given.

To simulate the sequence likelihood for trial $n$, we used the instruction sequences up to trial $n-1$. This allowed the HVM to learn in an online manner, i.e. abstractions and chunks were proposed upon each parsing completion. For comparison purposes, we also simulated HCM with the same parameter setting, and an associative learning model (AL) that updates first order transitions between atomic sequential units. The sequence likelihood was evaluated as $-\log P(S) = -\log P(c_1) \prod_{i=2,...,n} P(c_i|c_{i-1})$. $c_1, c_2, ..., c_n$ are the units used to parse the sequence (chunks for HCM and HVM and atomic units for the associative learning (AL) model).

Figure 4 shows the relation between model sequence likelihood and human recall time during the training (b, c) and the transfer block (d, e). The size of the dot represents the trial number, and the shape of the dot represents the group (the triangle being the variable group, the circle is the control group). Figure 4b and d show the R-squared goodness-of-fit regressing the various models' sequence likelihoods onto human subjects' sequence recall times. The sequence negative log-likelihood from the HVM correlates with human recall time during the training ($R = 0.86, p \leq 0.001$) and transfer

blocks ($R = 0.7, p \leq 0.001$). The latter suggests that HVM can generate behavior that resembles that of human knowledge transfer in the memorization of novel sequences with embedded variables.

## 4.4 COMPARING ABSTRACTION LEARNING AND TRANSFER IN LLMS

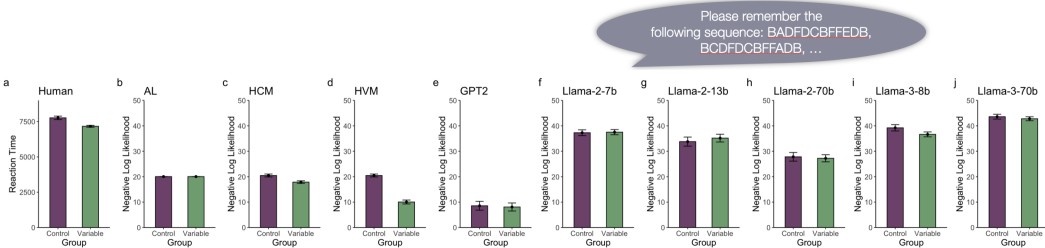

Figure 5: Comparison between human, variations of cognitive models, and AI models on the transfer block of the memory recall experiment. Bar plot shows the average human sequence recall time, and the sequence negative log-likelihood evaluated by the various cognitive and large language models.

In recent years, large language models (LLMs) have demonstrated their emergent abilities (Wei et al., 2022), raising the concern that their intelligence will soon exceed that of humans, while at the same time, LLMs exert severe limitations in a set of tasks that demands abstract thought (Fleuret et al., 2011; Odouard & Mitchell, 2022). Given the critical role that abstraction plays in reasoning and generalization, it becomes increasingly important to delineate the aspects of LLM that most significantly deviate from human cognition, and find out what features inside a cognitive model would provide a missing behavioral aspect inside LLMs. The missing feature will inform the limitations of LLM's computational component that is predictive of its ability to generalize and transfer. Specifically, we ask whether LLMs abstract similarly to humans and how their behavior compares to cognitive models incorporating abstraction. To explore this question, we conduct the same sequence recall experiment on LLMs.

We adapt the instructions from the human experiment into prompts for in-context learning in LLMs. The LLMs are tasked with predicting the next subsequent tokens based on the context of previously seen instruction sequences. The specific prompt is taken from the instruction given to the human participants: "Please remember the following sequence: BADFDCBFFEDB, ..., BCDFDCBFFADB." We then calculate the conditional probability of the next token, $z_i$, given the instruction sequence history up until the current token $P(z_i|prompt)$. After obtaining this probability, the subsequent token in the instruction is added to the prompt, and the LLM's conditional probability for the following token is updated accordingly. Using this prompt-chaining method, we can calculate LLM's negative log-likelihood for the next tokenized instruction sequence: $-\log P(S) = -\log P(z_1|prompt) \prod_{i=2,...,n} P(z_i|prompt_{i-1})$, analogous to the cognitive models. We apply this approach to evaluate the token prediction likelihood in four large language models: GPT-2 (Radford et al., 2019) and three variations of the Llama 2 model (Touvron et al., 2023) and two variations of the Llama 3 model.

Shown in Figure 5 are the transfer behavior across humans, LLMs, and various cognitive models for comparison. Supplementary section A.11 also shows the R-square value regressing model negative log-likelihoods on human recall time. For humans, the group trained on sequences with embedded variables recalls transfer sequences faster than the group trained on control sequences. Relating sequence recall time with sequence negative log-likelihood, the cognitive models HCM and HVM, on average, exhibit lower sequence negative log-likelihood after learning from a training block that shares variables with the transfer block. In contrast, the associative learning model and all variants of large language models do not differentiate between training sequences that share variables with the transfer block and those from a training block without transferrable variables.

## 4.5 ABSTRACTION, DISTORTION, AND GENERALIZATION

With a model that can learn interpretable abstraction, we can delve into the relationship between learning layers of abstraction, compression, and uncertainty. HVM updates its dictionary each iteration

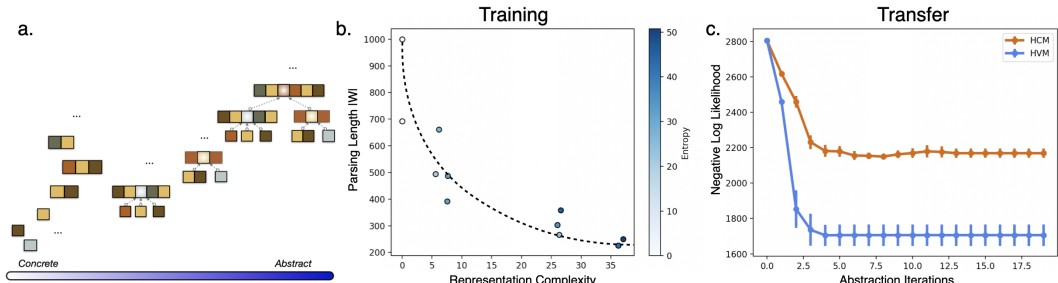

Figure 6: a. New chunks or variables are created from previously learned ones, adding increasingly nested structures to the representation graph. b. As the allowable distortion increases (moving right on the x-axis), the required rate to encode the data in terms of chunks decreases (moving down on the y-axis). Introducing variables enables a more compressed sequence, which trades off with higher representational complexity introduced by the variables embedded inside the chunks. c. Abstraction and transfer: The higher the abstraction layer, the higher the likelihood (lower negative log-likelihood) that HVM parses novel sequences.

by proposing chunks and variables, which are used to parse the sequence for the next iteration. New chunks or variables are created from previously learned ones, adding increasingly nested structures to the representation graph. This feature affords a controlled assessment of the relation between the level of abstraction, the amount of distortion that abstract representation introduces, and its relation to transfer and generalization to novel sequences. We measure nestedness using the representation complexity $RC(\mathbb{V}) = \sum_{v \in \mathbb{V}} \sum_{u \in \mathbb{E}(v)} -\log P(u|v)$, which is closely related to the encoding cost of the representation graph A.5. We also measured the uncertainty carried by the embedded variables in the chunks via entropy formulated as $\sum_{c \in \mathbb{C}} P(c) \sum_{v \in V(c)} \sum_{u \in E(v)} -P(u|v) \log P(u|v)$. Figure 6a shows the relation between the three factors as a consequence of increasing abstraction learning iterations. As the representation graph becomes increasingly nested, the newly learned chunks explain a longer part of the sequence, and these chunks embed more uncertain variables. This trade-off between compression and uncertainty reflects rate-distortion theory specifying that the best possible compression of a signal $X$ contains a lower bound on quality loss specified by the Rate-Distortion Function (R(D)): $R(D) = \inf_Q R_Q \text{s.t.} \mathbb{E}[d(X, \hat{X})] \leq D$ (Shannon, 1959; Cover & Thomas, 2012). Taking representation complexity as a distortion function, as the level of abstraction increases, HVM learns patterns that span and predict longer parts of the sequence while more distortion and uncertainty are introduced. This observation on distortion with the level of abstraction resonates with previous findings relating mental errors with learning more abstract representations in humans (Lynn et al., 2020).

**Higher levels of abstraction implies more flexible transfer** If learning higher abstraction layers implies more distortion, what would be the benefit apart from compression? We suggest that the other side of the coin lies in generalization and demonstrates the effect of abstraction on model transfer. We trained HVMs with increasing abstraction layers and evaluated the models' transfer likelihood upon parsing a novel sequence. As shown in Figure 6b, the higher the abstraction layer, the higher the likelihood (lower negative log-likelihood) that HVM parses the transfer sequence. To compare, we also evaluated the transfer performance of the chunking model (HCM) with an increasing level of abstraction iteration. While the transfer performance for HCM improves with more layers of chunk learning, it converges at a higher negative log-likelihood. Learning an increasing abstract representation by the HVM enables it to compress novel transfer sequences in a more succinct way. Furthermore, the more abstraction layers are acquired, the higher the relative advantage for HVM that learns abstraction holds over the HCM that does not learn variables.

## 5 RELATED WORK

Previous modeling work on abstraction can be divided into two categories. The first is cognitive models, which characterize abstraction as searching for commonalities in explicitly symbolic systems (Lu et al., 2021); these models apply to explain human problem solving, finding problem solutions

based on its conceptual analogies with the previously-solved problems (Hofstadter & Mitchell, 1994). These models describe knowledge using symbolic, explicit graphs but do not address how the representation arises from perceptual data. Other models similar in taste include the recent wake-sleep Bayesian models such as DreamCoder Ellis et al. (2021). Our work proposes a mechanism for learning explicit abstraction from sequences. HVM starts learning with an empty dictionary and does not assume program primitives. Additionally, the inference/recognition in the HVM is sufficiently simple and can be simple without extensive machinery.

The second model category implements abstraction in implicit connectionist systems. Works on meta-learning models and LLMs suggest their ability to generalize across contexts and solve problems in a way similar to humans in some tasks (Wei et al., 2022; Binz & Schulz, 2023) while failing short on other abstract reasoning tasks (Fleuret et al., 2011). Meanwhile, abstraction has been argued to be implicitly present in artificial neural networks (Yee, 2019; Kozma et al., 2018; Johnston & Fusi, 2023; Ito et al., 2022) and biological neural activities (Bernardi et al., 2020; Goudar et al., 2023), albeit challenging to interpret (Fan et al., 2021). Relative to this approach, our work provides a reductionist cognitive model that explicitly specifies the minimal components needed to learn interpretable abstract structures.

Besides cognitive models, our work relates to and differentiates from other sequence learning approaches. Our generative mechanism relies on chunk-based recursive generation and inventory growth rather than formal grammar rules (Jelinek et al., 1992; Johnson et al., 2006), Zipfian distributions (Teh, 2006), abruptly changing statistics (Milan et al.), or sequences from a stochastic chain Willems et al. (1995). We suggest that the perceptual sequence contains nested hierarchical structures, with chunks as a central generative force.

The hierarchical nature of HVM shares with models including topics models and Helmholtz Machine; however, it differs from topics models Blei et al. (2003); Teh et al. (2006) aiming at sequences rather than 'bags of words'. The Helmholtz Machine also contains hierarchical generative and recognition components, but its focus is similarly not on sequences, and its sequential extension Hinton et al. (1995) did not work well partly because a lack of iterative processing that HVM employs. Finally, the goal of HVM as a computational cognitive model differs from neural models such as CSCG (George et al., 2021) which has an implementation level focus Marr (1982). However, we encourage future work to investigate the neural underpinnings of abstraction.

## 6 DISCUSSION

Our work has limitations. One is that variables are only proposed to be embedded between chunks and cannot come at the beginning or end of sequences, restricting the location of discoverable variables. Secondly, representation learned later in iteration depends on those earlier acquired. However, this aspect aligns with the order and curriculum dependency of human learning. Children demand a simple-to-complex curriculum and utter more complex compounds later in their development (Friedmann et al., 2021; Snow, 1972; Fernald et al., 1989). There is a limit to HVM's expressivity: it won't be able to capture a pattern of two chunks being a variable number of chunks apart. Sometimes, the variable structure can be arbitrarily nested depending on the representations learned up until that moment. Future work may look at optimizing the structure during learning. Finally, our work focused on the cognitive resemblance of HVM and has yet to explicitly optimize the model for computational efficiency, leaving space for future work.

Our work opens up future directions both in cognitive science and machine learning. Previously, grammar learning (Chomsky, 2013), chunk learning (Gobet et al., 2001), and statistical/associative learning (Saffran et al., 1999) were studied as distinct characteristics of human sequence learning. We propose via HVM that the three components can be treated as finding invariant recurring chunks from sequences via statistical/associative learning, resulting in learning grammar-like structures. Our work suggests a normative origin of concrete and abstract chunk learning as a learning agent uncovers the underlying entities that constitute perceptual sequences. Future work can relate this model to developing concepts or compare the model with human behavior and LLMs on more nested datasets. Meanwhile, the close tie between abstraction and generalization also urges future hypothesis-driven research regarding preconditions for learning abstract concepts to acquire more complex representations. This can help to illuminate the emergence of abstract representation through excessive training and its relation to the generalizability of connectionist models (Power et al., 2022;

Miller et al., 2024). Finally, our work can act in the tokenization role to fuse with transformers or SSM to extract interpretable representations from complex sequence data while enjoying the power of history-dependent sequence prediction.

## 7 Reproducibility Statement

Detailed information about the HVM algorithm, proof, generative model, test and experimental details and results can be found in the supplementary information section. The code used for the algorithm and experiments is available under this link.

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

# A    APPENDIX / SUPPLEMENTAL MATERIAL

## A.1    SET UP

An observational sequence $S$ is made up of discrete, integer valued, size-one elementary observational unit coming from an atomic alphabet set $\mathbb{A}$. One example of such an observational sequence $S$ is:

$$010021002112000...$$

An example belief set that contains only concrete chunks can be $\mathbb{B} = \{0, 1, 21, 211, 12, 2112\}$.

Using the belief set to parse the sequence $S$ results in the following partition. $\underline{0}\ \underline{1}\ \underline{0}\ \underline{0}\ \underline{21}\ \underline{0}\ \underline{0}\ \underline{2112}\ \underline{0}\ \underline{0}$
$\underline{0}$.

Another example belief set $\mathbb{B}$ that contains chunks $\mathbb{C} = \{21v, 0100, 000\}$ with embedded variables $\mathbb{V} = \{v\}$. The denoting set of $E(v) = \{00, 12\}$. Then the sequence $S$ is parsed as $\underline{0100}\ \underline{21v}\ \underline{21v}$
$\underline{000}$.

**Definition 1 (*Completeness*)**

*We say that a belief set is* complete *if at any point during the sequence parsing process, the upcoming observations can be explained by at least one chunk in the belief set.*

In this work, the learning mechanism guarantees that the belief set is complete.

**Definition 2 (*Parsing Length $|W|$*)**

*A parsing length $|W|$ of a sequence is the length of the sequence measured in chunks.*

## A.2    GENERATIVE MODEL

---

**Algorithm 1:** Pseudocode to generate sequences with nested abstract hierarchies.

---

**Input:** A set of atomic elements $\mathbb{A}$; the number of combinations $d$; sequence length $l$
**Output:** $seq$, a sequence made of concrete observational units
$cg \leftarrow$ initialize representation graph;
**for** $i \leftarrow 1$ **to** $d$ **do**
    $RAND \leftarrow$ random number between 0 and 1;
    **if** $RAND > 0.5$ **then**
        `// Object Creation`
        $B \leftarrow cg.\text{objectsAndCategories}$;
        $n_{combo} \leftarrow \text{random.choice}([2, 3, 4, 5])$;
        $samples \leftarrow \text{random.sample}(B, \text{k=}n_{combo})$;
        **while** *any of the first and last element in $samples$ are categories* **do**
            $samples \leftarrow \text{random.sample}(B, \text{k=}n_{combo})$;
        **end**
        $newobject \leftarrow \text{concatenate}(samples)$;
        $cg.\text{addChunk}(newobject)$;
    **end**
    **else**
        `// Category Creation`
        $B \leftarrow cg.\text{objects}$;
        $n_{combo} \leftarrow \text{random.choice}([2, 3, 4, 5])$;
        $samples \leftarrow n_{combo}\ \text{random.sample}(B, \text{k=}n_{combo})$;
        $newcategory \leftarrow$ create a new category denoting $samples$;
        $cg.\text{addVariable}(newcategory)$;
    **end**
**end**
$cg.\text{assignProbabilitiesToObjects}()$;
$sampledseq \leftarrow cg.\text{sampleObjectAndSpecifyVariables}(l)$;
$seq \leftarrow$ convert $sampledseq$ to sequence;

---

A new object is created by concatenating a random selection of pre-existing objects or categories from the existing inventory. After the inventory has been expanded up to the $d$-th iteration, objects are assigned with an independent occurrence probability sampled from a flat Dirichlet distribution $f(p(c_1), ..., p(c_{|\mathbb{A}|}); \alpha_1, ..., \alpha_{|\mathbb{A}|}) = \frac{1}{\mathbf{B(a)}} \prod_{i=1}^{|\mathbb{A}|} P(c_i)^{\alpha_i - 1}$, $\alpha_i = 1 \forall i$. Where the beta function when expressed using gamma function is: $\mathbf{B(a)} = \frac{\prod_{i=1}^{|\mathbb{A}|} \Gamma(\alpha_i)}{\Gamma(\sum_i^{|\mathbb{A}|} \alpha_i)}$, and $\mathbf{a} = (\alpha_1, .., \alpha_{|\mathbb{A}|})$. The parameters $(\alpha_1, .., \alpha_{|\mathbb{A}|})$ are identically set to one.

Similarly, a probability is sampled from a flat Dirichlet to assign the independent occurrence probability of the set of object $E(v)$ that each category $v$ denotes. $\forall v \in \mathbb{V}$, $f(p(c_1), ..., p(c_{|E(v)|}); \alpha_1, ..., \alpha_{|E(v)|}) = \frac{1}{\mathbf{B(a)}} \prod_{i=1}^{|E(v)|} P(c_i)^{\alpha_i - 1}$, $\alpha_i = 1 \forall i$. This procedure is done for each created category.

Intuitively, objects represent recurring observations of specific entities, such as molecules composed of chemical elements. Categories, on the other hand, represent broader categories of entities that share common properties, such as chemical elements belonging to the same class (e.g., noble gases or alkali metals). The observation sequence forms a nested hierarchy, where entities and their categories frequently interact and combine with others.

### A.3    APPROXIMATE RECOGNITION INVERSE

**Theorem 1.** *If the left entity $c_L$ and the right $c_R$ (which can be either a chunk or a variable) of a ground truth variable $v$ in addition to its denoting chunks $c_1, c_2, ..., c_m$ has been learned, and every parsing of $c_L$, $c_R$, and $c_1, c_2, ..., c_m$ is consistent with the constitution of the sequence by the its way of generation, then $c_1, c_2, ..., c_m$ will be necessarily proposed by the common adjacency and common preadjacency criterion into a novel variable $v'$ and the true denoting entities will be a subset of the denoting set $E(v')$, $\{c_1, c_2, ..., c_m\} \in E(v')$.*

*Proof.* By definition, $\{c_1, c_2, ..., c_m\}$ is in the adjacency $Adj(c_L)$ entries of $c_L$ and the preadjacency entries of $c_R$, $Preadj(c_R)$. And therefore, $\{c_1, c_2, ..., c_m\} \in Adj(c_L) \cap Preadj(c_R)$ and hence will be included in the set of denoting entities for a new variable. $\square$

## Proposing variables based on transition matrix

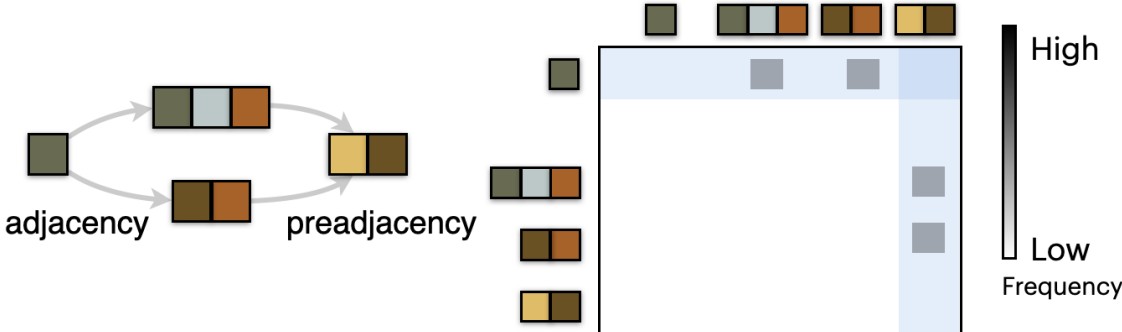

Figure 7: Common adjacency and preadjacency structure to identify a variable.

**Theorem 2.** *In an infinitely long sequence, a chunk $c$ in the generative model is made up by concatinating the entities $\{c_1, c_2, ..., c_n\}$, and these entities $\{c_1, c_2, ..., c_n\}$ are parsed in an identical way as how the generative model samples $c$, then HVM will eventually learn to chunk $c'$ including all the entities in $\{c_1, c_2, ..., c_n\}$.*

*Proof.* By contradiction. If any of the entities are not included in the learned chunks, a correlation will still exist in the transition entries between subparts that compose $c$, which will be resolved by the next chunk proposal iteration. $\square$

In practice, identifying the ground truth representation graph is much more complicated, as the representations learned by HVM can be mapped to a context-sensitive grammar A.6, which is more complex and undecidable than context-free grammar. Learning the correct chunks that exactly match the generative model relates to the non-identifiability problem that multiple grammars can generate the same set of observed data (Post, 1946; Greibach, 1968) - inferring the exact grammar from positive examples is undecidable. The sequence length comparable to the ground truth is possible to obtain. Instead of comparing the model with the specific chunks of the ground truth, we use a set of evaluation measures to evaluate the model's performance.

---

**Algorithm 2:** HVM (online version, for learning sequences from human experiments)

---

**Input:** Learning sequences $seq$; representation graph $cg$; boolean flag for chunk learning $threshold\_chunk$; boolean flag for variable learning $abstraction$

**Output:** $cg, chunk\_record$

$cg \leftarrow$ initialize representation graph;
// Initialize chunk record
$chunk\_record \leftarrow \{\}$;
$t \leftarrow 0$;
**while** *not seq_over* **do**
  $current\_chunks, cg, seq, chunk\_record \leftarrow identify\_latest\_chunks(cg, seq)$;
  $cg \leftarrow learning\_and\_update(current\_chunk, chunk\_record, cg, threshold\_chunk = True)$ ;
  **if** $abstraction$ **then**
    $cg \leftarrow abstraction\_update(current\_chunks, cg)$;
  **end**
  $cg.forget()$;
  // multiply all frequency record by $\theta$
**end**

---

A.4   PARSING SEARCH STEPS

HCM retrieves learned chunks to match them with the upcoming sequence, parsing the largest chunk that aligns with the sequence. However, as the dictionary size $|\mathbb{C}|$ grows, the number of search steps increases, leading to inefficiency. To address this, HVM organizes the chunks in $\mathbb{C}$ within a parsing graph, harnessing shared nodes across multiple chunks to speed up the chunk identification process.

**Definition 3 (*Parsing Graph (PG)*)**

*Parsing graph is a graph for chunk identification. The nodes inside such a graph are the chunks $\mathbb{C}$. Each parent node is the overlap of all of its children.*

To parse a chunk, the model traverses the memory tree structure to find the path up to the deepest node inside the tree consistent with the upcoming sequence. Pseudocode A.4 describes this process.

---

**Algorithm 3:** Pseudocode for traversing a tree to find a path consistent with an upcoming sequence

---

**Input:** $rootNode$, the root node of the tree; $sequence$, the sequence to be matched
**Output:** $path$, the path to the leaf consistent with the sequence, if found
$path \leftarrow$ an empty list;
**if** *not rootNode* **then**
   | **return** $null$;
**end**
**else**
   | **return** TRAVERSETREE(()$rootNode$, $sequence$, $path$);
**end**
**Function** `TraverseTree` (*node, sequence, path*) **:**
   | $path$.append($node$);
   | **if** *IsConsistent(node, sequence)* **then**
   |    | **if** $node.isLeaf()$ **then**
   |    |    | **return** $path$;
   |    | **end**
   |    | **forall** $child$ $in$ $node.children$ **do**
   |    |    | $result \leftarrow$ TRAVERSETREE($child$, $sequence$, $path$.copy());
   |    |    | **if** $result$ **then**
   |    |    |    | **return** $result$;
   |    |    | **end**
   |    | **end**
   | **end**
   | **return** $null$;

---

**Definition 4 (*Parsing Search Steps*)**

*Parsing search step refer to the number of search comparison made in the parsing graph to find the deepest chunk in the tree consistent with the upcoming sequence.*

**Example** Shown in Figure 8, when the sequence upcoming is 123, one starts from the ancestors of the graph and check whether 1 or 2 is at the beginning of the sequence, identifies 1, and proceeds into the subtree which stems from 1. The leaves are contains 12 and 13. Then HVM checks whether 12 is in the sequence or 13, thereby identifying 12.

The mechanism to identify the biggest concrete chunk that can be used to parse the sequence involves the following steps:

- Search from the top of the tree (nodes with no parents)
- Find the node in the next layer consistent with sequence
- Go to the children of the node until the node contains no more children (leaf node) or none of the node's children is consistent with the sequence

As illustrated in Figure 8, the parsing search steps of chunk 12 is: $PSS(12) = 2 + 2 = |A| + |R(A)|$
For chunk 221: $PSS(221) = 2 + 2 + 3$ For 21: $PSS(21) = 2 + 2$

**The advantage of memory abstraction**

**Theorem 3** (Guarantee on fewer search steps). *Identifying a chunk in a tree-structured memory takes fewer search steps than without.*

*Proof.* The hierarchically structured memory system reduces the searching step to the number of branches on the path that leads to the identified chunk, which is a subset of all nodes in the tree. □

**Expected parsing search steps** For a chunk $c$ that occur with probability $P(c)$, the parsing search steps is denoted as $PSS(c)$.

As an example, three chunks $c_1$, $c_2$, and $c_3$ composes a sequence and they are parsed with probability $P(c_1)$, $P(c_2)$, and $P(c_3)$ in a sequence.

Figure 8: Example process when parsing a chunk using parsing graph. Gray region denotes chunk overlaps, which are the non-leaf nodes inside the parsing graph being the common prefix of their children.

Without the hierarchical memory structure, the average parsing search steps to identify chunks in the sequence is:

$$PSS(c_1)P(c_1) + PSS(c_2)P(c_2) + PSS(c_3)P(c_3)$$

With the prefix Trie, the number of times needed to check whether a chunk matches a certain abstraction branch requires a number of parsing steps:

$$PSS(c) = N_{children}(Pa(c)) \tag{1}$$

$Pa(c)$ denotes the parent of chunk $c$ in $PG$, $N_{children}(Pa(c))$ is the number of children that chunk $c$'s parent contain. An abstract chunk $a$ in such a parsing graph can be the prefix intersection of three chunks $c_1$, $c_2$, and $c_3$, $a = c_1 \cap c_2 \cap c_3$. Parsing abstraction summarizes all the subordinate chunks that underlies such abstraction. Therefore the abstraction node is more likely to be parsed compared to individual chunks.

$$P(a) = (P(c_1) + P(c_2) + P(c_3))$$

Additionally, abstraction can be used as an anchor to find the subsequent denoting chunks. $P(c_1|a)$, $P(c_2|a)$ and $P(c_3|a)$. The expected parsing search steps to identify chunks for such sequence $S$ becomes:

$$\mathbb{E}_S(PSS) = PL(c_1 \cap c_2 \cap c_3)(P(c_1) + P(c_2) + P(c_3))$$
$$+ PL(c_1 - a)P(c_1|a) + PL(c_2 - a)P(c_2|a) + PL(c_3 - a)P(c_3|a)$$

Generalizing this formula to any parsing graph, the expected parsing search steps given the parsing graph's internal abstraction and concrete chunks would be

$$EPSS(PG) = \sum_{c \in \mathbb{C}} P(c) \sum_{path(c)} len(N_{children}(PA(c))) \tag{2}$$

$path(c)$ is the path starting with the root node of the graph that leads to the identification of the chunk node c. Since $P(c)_{c_i \in path(c)} = P(a) \prod_{c_i \in path(c)} P(c_i|pa(c_i))$, one can also give a lower and upper bound on the number of parsing search steps needed to arrive at a random chunk in the parsing graph:

$$EPSS(PG) = \sum_{v \in \mathbb{C}} P(v|PA(v))PSS(v - PA(v)) \tag{3}$$

$PSS(a - b)$ denotes the subgraph that starts from $b$ which reaches $a$. If $v$ is inside the ancestor nodes, then $ancestor(v) = \varnothing$, and therefore $P(v|ancestor(v)) = P(v)$.

The expected parsing search steps for a representation graph can be calculated by averaging out the expected parsing search steps for each sub graph inside the parsing graph. The lower bound is the number of steps that leads to the shallowest node, and the upper bound is the number of steps that leads to the deepest leaf node.

$$\arg\min_{c \in \mathbb{C}} \sum_{u \in path(c)} len(n_{children}(PA(c))) \le \mathbb{E}_{c \in \mathbb{C}}(c) \le \arg\max_{c \in \mathbb{C}} \sum_{u \in path(c)} len(n_{children}(PA(c)))$$
$$\tag{4}$$

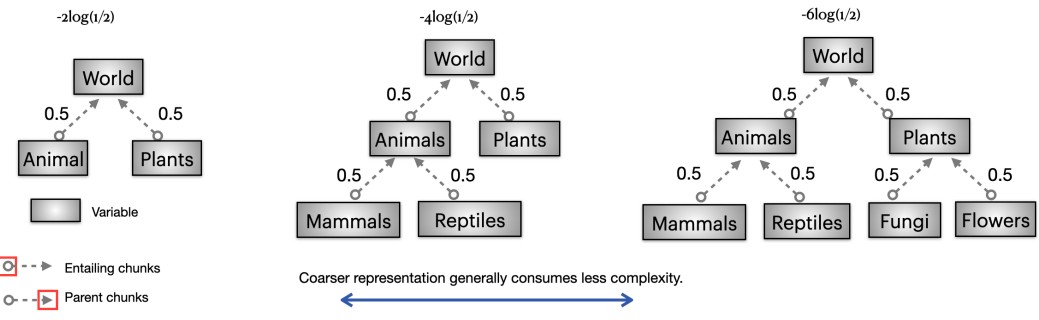

Figure 9: An increasing level of abstractions

## A.5 ENCODING COST

A representation graph $RG$ specifies a distribution of observation instances encoded in a compositional manner. One can the minimal encoding cost to distinguish all the variables.

$$RC(G) = \sum_{v \in G} \sum_{u \in E(v)} -\log P(u|v) + \sum_{v \in G} -\log P(v) + \sum_{c \in G} -\log P(c) \tag{5}$$

The more ambiguous a variable is, the less resources needed to encode the variable; the smaller the variable graph is, the less edges it has, the bigger the probabilities of parsing and variable identification, the smaller the encoding cost. Every time when a new edge points from a pre-existing variable to a new variable, the encoding resource expands by an amount determined by the conditional probability.

**Example** Representation graphs with different nested structure translates to different encoding cost. In Figure 9, three graphs contains an increasing abstraction specificity. In graph 1, the variable 'world' is split into two variables: animals and plants. Each of these variables do not denote a specific observation, but serves as a placeholder for any instances of specific observation that fits into that particular category. The encoding cost for such a graph, given the specified observational probability on the edges would be:

$$EC(g1) = -\log(P(World)) - \log(P(Plants|World)) - \log(P(Animal|World)) = -2\log(0.5) \tag{6}$$

Since once the variable $World$ has been identified, one only need to distinguish the subcategories inside the variable $World$, and the minimal code length to distinguish one subcategory (animals) from another (plants) would be the conditional probability: $P(Plants|World)$.

Whereas if $g1$ does not contain a nested structure, one would need to encode the variable world separately from the variable animal and variable plants, in this way, the alternative encoding length without an edge connecting the sub categories with the main category would be:

$$EC(\hat{g1}) = -\log(P(World)) - \log(P(Plants)) - \log(P(Animals)) = -2\log(0.5) \tag{7}$$

For $g2$, the case is slightly different: the variable category that denotes animals is further split into the category of mammals and reptiles. Thereby, extra encoding costs are needed.

$$\begin{aligned} EC(g2) = &-\log(P(World)) - \log(P(Plants|World)) - \log(P(Animals|World)) \\ &-\log(P(Mammals|Animals)) - \log(P(Reptiles|Animals)) \\ &= -4\log(0.5) \end{aligned} \tag{8}$$

If each of these categories are encoded separately and each variable is not pointed to other variables, the information content to encode a variable graph without edges in the case of $\hat{g2}$ would be:

$$EC(\hat{g}2) = -\log(P(World)) - \log(P(Plants))$$
$$= -\log(P(Animals)) - \log(P(Mammals)) - \log(P(Reptiles)) \quad (9)$$
$$= -6\log(0.5)$$

Since $P(Mammals) = P(mammals|animals)P(animals)$.

This difference is more pronounced going from $g2$ to $g3$:

$$EC(g3) = -\log(P(World)) - \log(P(Plants|World)) - \log(P(Animals|World))$$
$$- \log(P(Mammals|Animals)) - \log(P(Reptiles|Animals))$$
$$- \log(P(Fungi|Plants)) - \log(P(Flowers|Plants)) \quad (10)$$
$$= -6\log(0.5)$$

Whereas for $\hat{g}3$, not encoding variables in a nested structure would result in an encoding cost of

$$EC(\hat{g}3) = EC(\hat{g}2) + -\log(P(Fungi)) - \log(P(Flowers))$$
$$= -10\log(0.5) \quad (11)$$

These examples illustrate organizing variables into a nested recursive structure saves encoding cost.

### A.6 Alternative formation as learning a context sensitive grammar

HVM learns a 5-tuple from the sequences $G = \{\mathbb{A}, \mathbb{C}, \mathbb{V}, \mathbb{R}, \mathbb{P}\}$. A set of atomic units $\mathbb{A} = \{a_l\}$
A set of chunks $\mathbb{C} = \{c_k\}$, $k = 1, 2, ...,$, each chunk $c_k$ is a sequence of atomic units and variables. The model uses chunks from $\mathbb{C}$ to parse the observation sequence. Assume a random variable $C$ as the chunk being parsed, taking a value $c$ from $\mathbb{C}$, $c \in \mathbb{C}$, $c \sim P$. $P$ is the parsing probability. $\sum_{c \in \mathbb{C}} P(C = c) = 1$
A set of variables $\mathbb{V} = \{v_i\}$, $i = 1, 2, ...,$
A set of rules $\mathbb{R} = \{E(v_1), E(v_2), ..., E(v_n)\}$ that each specifies the set of chunks $E(v) = \{c_j\}$ denoting each variable $v \in \mathbb{V}$. Probabilities on variable denoting $\forall i, \sum_j P(v_i \to c_j) = 1$

### A.7 Rate-Distortion Theory

Rate-distortion theory specifies that the best possible compression of a signal $X$ contains a lower bound on quality loss specified by the Rate-Distortion Function (R(D)): $R(D) = \inf_Q R_Q \text{s.t.} \mathbb{E}[d(X, \hat{X})] \leq D$ (Shannon, 1959; Cover & Thomas, 2012). According to the $RD$ theory, the minimum $R$ (the amount of compression) at which information can be transmitted over a communication channel for a given level of information loss D is specified by the **Rate-Distortion Function** $(R(D))$: $R(D) = \min_{p(\hat{x}|x):E[d(X,\hat{X})] \leq D} I(X; \hat{X})$. The mutual information quantifies how much information needs to be preserved during compression to achieve this fidelity, and the function minimizes this quantity while satisfying the distortion constraint. Written in another way, $R(D) = \inf_Q R_Q$ s.t. $E[d(X, \hat{X})] \leq D$. $Q$ represents the encoding function that maps from $x$ to $\hat{x}$. RD is agnostic to the choice of the distortion function, and in this case we use the representation complexity, which measures the nestedness of the variables learned by HVM.

### A.8 Parsing Probability

A sequence $S$ is parsed by the chunks in $\mathbb{C}$ to obtain a distribution of chunk parsing probability $P_{\mathbb{C}}$. This distribution on the support set of chunks has shapes dependent on the parsing mechanism. HVM employs a greedy strategy and chooses the deepest consistent chunk in the memory tree to parse a segment of the upcoming sequence. Other parsing strategies will result in alternative distributions for $P_{\mathbb{C}}$, in addition to the transition probability $P_{\mathbb{C}}(c_i|c_j)$.

Define the set $H_{\mathbb{C}}(S)$ as the set of all distributions results in any parsing method using the chunks in $\mathbb{C}$ to parse a sequence $S$, and any parsing probability $P \in H_{\mathbb{C}}(S)$. Hence each evaluation measure on

the representation is bounded by the optimal and the most unfortunate parsing occasions. Taking the predictive power measure as an example:

$$\arg \min_{P \in H_\mathbb{C}(S)} \mathbb{E}_P |c| \leq \mathbb{E}_{P_{HVM}} |c| \leq \arg \max_{P \in H_\mathbb{C}(S)} \mathbb{E}_P |c| \tag{12}$$

The measured value is bounded by the expected value from the best parsing distribution and the worst parsing distribution.

### A.9 UNCERTAINTY

Variables introduce uncertainty. Without variables, the chunks inside $\mathbb{E}_v$ needs to be encoded as distinct chunks, consuming an encoding cost of $\sum_{c \in \mathbb{E}_v} - \log P(c)$.

We have the ground truth set of sequential observational units $gt$. Let's say in a recall task, the agent learns a variable chunk to describe the ground truth, the variable denotes to three concrete chunks $c_1$, $c_2$ and $c_3$. The accuracy when $c_1$ is being sampled will be $D(gt, c_1)$, when $D$ is a distance function evaluating the agreement between the ground truth chunk $gt$ and $c_1$. Then one can evaluate the expected accuracy within a variable:

$$\mathbb{E}(gt, v) = \sum_{c \in \mathbb{E}_v} P(c|v) D(gt, c) \tag{13}$$

$P(c|v)$ is the probability of sampling chunk $c$ given that the variable $v$ is identified.

Compared to chunks without variables, recalling chunks with embedded variables introduces variability, and brings down recall accuracy. But this strategy is especially suited when encoding resources is limited. Then, it is better when the resources are assigned to the more probable chunks that are predictive of a bigger sequence, while the varying entities that occur with lower probability can be denoted as a variable that serves as a placeholder.

$$Uncertainty(G) = \sum_{c \in \mathbb{C}} P(c) \sum_{v \in C} H(v) \tag{14}$$

And $H(v) = -\sum_{c \in E(v)} P(c) \log P(c)$.

### A.10 OTHER EVALUATION MEASURES

**Explanatory Volume:** $\frac{|S|}{|W|}$ The length of the original sequence (in atomic units) divided by the length of the parsed sequence (in units of chunks), i.e., The average size of the sequence that the current representation graph can explain at each parsing step.
**Sequence Negative Log Likelihood:** The lower bound of information content needed to encode the sequence $S$, as $S$ is parsed into chunks $(c_1, c_2, ..., c_n)$, $-\log P(S) = -\log(\prod_{c_i \in \mathbb{C}} P(c_i))$.
**Representation Entropy:** The uncertainty contained within each chunk parse $\sum_{c \in \mathbb{C}} P(c) \sum_{v \in V(c)} \sum_{u \in E(v)} -P(u|v) \log P(u|v)$

### A.11 REGRESSING LLM ON HUMAN DATA

We also regressed the negative log likelihood (NLL) of all LLMs against human sequence recall time and presented the resulting R-squared values in Figure 10. During the training block, the NLL of LLMs shows a stronger correlation with human recall times compared to alternative models. However, when it comes to human transfer, the cognitive models demonstrate a much stronger correlation than the LLMs, with HVM aligning most closely to human transfer performance.

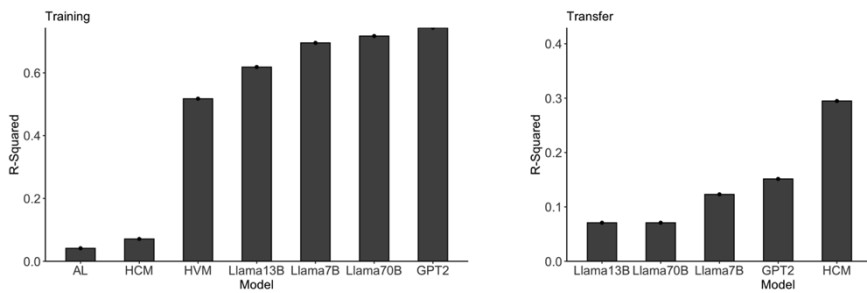

Figure 10: Regressing all models' sequence likelihood on human sequence recall time.

