# OpenReview forum: "Building, Reusing, and Generalizing Abstract Representations from Concrete Sequences"
_ICLR.cc/2025/Conference — ICLR 2025 Poster_

### Official Review · Reviewer_z6tL · 2024-11-02

**Soundness:** 3
**Presentation:** 3
**Contribution:** 3
**Rating:** 8
**Confidence:** 2

**Summary:**

The authors introduce a hierarchical variable learning model (HVM) that learns abstractions on sequence data via chunking and variable learning. Sequences are first parsed according to a parsing graph containing existing chunks hierarchically. If one chunk follows another chunk in a statistically significant manner, their concatenation is added to the list of chunks alongside hierarchical information. Variables are also identified as a set of chunks that consistently occur before/after another chunk. Variables can also be used in chunk formation.

The model is tested on both a synthetic dataset and real world sequential datasets (language modeling) and compared against HCM and LZ78 on parsing search steps, sequence length, sequence negative log-likelihood, and encoding efficiency. It is shown that the design of HVM does indeed provide benefits with respect to these metrics.

The authors also HVM reflects human memorization and transfer performance on a memorization and transfer task.

In addition, the authors compare HVM performance to LLMs and associative learning models on the same cognitive task.

It is also shown that higher levels of abstraction in HVM leads to relatively higher likelihood when parsing unseen sequences compared to HCM, suggesting improved generalization.

**Strengths:**

- The paper is organized and well written
- The paper includes evaluations of the model from many different perspectives, including evaluation on synthetic and real world language datasets, comparison with human performance on a cognitive task, etc
- The paper has a good level of technical rigour with the addition of definitions, theorems, and algorithms in the appendix.
- The topic of abstraction is of great significance to the field of AI, and the paper proposes a novel approach to tackling this issue

**Weaknesses:**

- Some small presentation issues in the appendix: equation in line 1020 is too long, figure 7 has a red line from a word processor, figure 8 has a red line that seems to be an error
- While I did praise the comprehensive evaluation, it is almost _too_ much content; it was difficult for me to understand the model without referring to the appendix. I suggest including information about the set up in the main text to improve clarity.
- Since the model builds up chunks by considering frequencies of adjacent chunks, there might be a limit to the expressivity of the formed abstraction. For example, while it might be able to capture the pattern of two chunks consistently being n chunks apart (if the intermediate chunks are all members of the set representing a variable), it cannot capture a pattern of two chunks being a variable number of chunks apart, where the number of chunks is determined by a symbol being present in the sequence.
- The evaluation only considers coding efficiency, compression efficiency, etc as a proxy for abstraction, which I do not believe to be sufficient for the purpose of demonstrating the effectiveness of the learned abstract representations. The demonstration of generalization does provide some evidence (Q1 below related to this point).

**Questions:**

- Would it be possible to use these learned representations for a task that requires abstraction (e.g. abstract reasoning)?
- When humans perform abstractions, semantic content also manners. Can this be integrated into the approach proposed in the paper?

---

> ### Author Response · Authors · 2024-11-23
>
> We sincerely thank Reviewer z6tL for the thoughtful and encouraging feedback. We are glad that the reviewer found the paper novel and well-written, with significant contributions to the field of abstraction in AI. Below, we address the specific weaknesses and questions raised.
>
> > Some small presentation issues in the appendix: equation in line 1020 is too long, figure 7 has a red line from a word processor, figure 8 has a red line that seems to be an error
>
> Thank you for pointing this out. We have corrected the equation and the figures.
>
> > While I did praise the comprehensive evaluation, it is almost too much content; it was difficult for me to understand the model without referring to the appendix. I suggest including information about the set up in the main text to improve clarity.
>
> Thank you for the suggestion, we have moved critical parts describing the model from the appendix to the main text.
>
> > Since the model builds up chunks by considering frequencies of adjacent chunks, there might be a limit to the expressivity of the formed abstraction. For example, while it might be able to capture the pattern of two chunks consistently being n chunks apart (if the intermediate chunks are all members of the set representing a variable), it cannot capture a pattern of two chunks being a variable number of chunks apart, where the number of chunks is determined by a symbol being present in the sequence.
>
> Thank you for the question. There is also a limit of HVM's expressivity: HVM will not be able to capture a pattern of two chunks being a variable number of chunks apart, indicated by a particular symbol embedded in the sequence. We have added this point in the limitation section (l 520).

---

> > ### Author Response · Authors · 2024-11-23
> >
> > > Would it be possible to use these learned representations for a task that requires abstraction (e.g. abstract reasoning)?
> >
> > Thank you for the comment. We are aware of dataset that evaluate abstraction by computing similarities between multiple-choice words based on text queries (BIG-bench, Srivastava et al. 2023), or designed with symbolic reasoning in mind (ARC challenge). As both benchmarks evaluate abstraction on more sophisticated settings, it is not readily adaptable to the simple sequence setting that we study in this paper, and the studies conducted on this benchmark uses large language models (Li et al., 2024). Currently, the algorithm that we propose contains the essence of abstraction in sequences but is still in a nascent stage to be plugged directly into abstraction tasks that are based on language description or symbolic reasoning. As this paper focuses on emphasizing the role of chunking in abstract pattern learning from sequences, we have taken this suggestion as a reference for future work to apply such models on benchmarks.  We mentioned this aspect in the discussion section of the paper to highlight as a future work.
> >
> > > When humans perform abstractions, semantic content also manners. Can this be integrated into the approach proposed in the paper?
> >
> > Thank you for the question. Indeed, the representation of abstraction is strongly related to the semantic content of the abstraction. Taking language as an example, the word ‘bank’ contains two semantic meanings: as a financial institution, or as a side of a river. People use semantic context to resolve the ambiguity and form an appropriate interpretation. Language models that only learn syntax-based abstraction struggles with this ambiguity (Spivey-Knowlton 1993, Frenck-Mestre 2000). However, a model that incorporates semantic understanding—such as recognizing relevant contextual cues—would be better equipped to interpret the sentence correctly.
> >
> > The incorporation of contextual cues comes automatically with the chunking component inside HVM. HVM groups elements into chunks or abstractions based on observed patterns and structures. The critical difference between HVM and syntactic parsing models is that context is critical to determine the variables. If the surrounding conversation is about finances or a busy city area, the word ‘bank’ would be chunked together with the context. In the case that the word ‘bank’ is included in some proposed variable, such as ‘economics’, the variable will be instantiated only when the contexts of the variable, (the financial/crowd chunks) are present. If the conversation is about nature or fishing, the word ‘bank’ would be chunked together with the context about nature or fishing. When parsing a new conversation, the context about fishing or nature shall elicit the fishing related chunks or variables that contain ‘bank’.
> > Therefore the word “bank” exists both inside finance related variables, and nature related variables, but conveys a different meaning and relates according to the context.
> >
> > Reference:
> > 1. Frenck-Mestre, C., & Pynte, J. (Eds.). (2000). Resolving syntactic ambiguities: Cross-linguistic differences? Cross-Linguistic Perspectives on Language Processing (pp. 119–148). Springer.
> > 2. Spivey-Knowlton, M. J., Trueswell, J. C., & Tanenhaus, M. K. (1993). Context effects in syntactic ambiguity resolution: Discourse and semantic influences in parsing reduced relative clauses. Canadian Journal of Experimental Psychology

---

> ### Comment · Reviewer_z6tL · 2024-11-26
>
> Thank you for addressing my concerns. I have no other questions and will be maintaining my score.

---

> > ### Author Response · Authors · 2024-12-02
> >
> > Thank you for your questions and constructive suggestions. We are grateful for your feedback which has helped us to improve this work.

---

### Official Review · Reviewer_dGcC · 2024-11-03

**Soundness:** 4
**Presentation:** 3
**Contribution:** 4
**Rating:** 6
**Confidence:** 4

**Summary:**

In this work, the authors build on a previous Hierarchical Chunking Model (HCM - a probabilistic model that learns to produce hierarchical chunks from sequential non-iid data) to create Hierarchical Variable learning Model (HVM), which groups chunks into high-level variable categories that have similar interaction properties. This model aims to compress sequences in a structured manner similar to humans. To test this out, the authors used sequence data from a variety of language datasets (childes, bnc, gutenberg, open subtitles) using HCM and LZW, a classical compression algorithm as baselines. The authors then used the model to account for human behavior in a sequence memory task that requires humans to re-use specific variables (against a control where there isn't a reusable variable). They also compared popular LLMs (GPT2, LLama2) on this task. HVM showed the biggest difference between the control and variable groups, which is the main effect that humans had.

**Strengths:**

* I think this is a really innovative algorithm that builds on the recent HCM in a pretty novel and cool way. It's clearly motivated by the human ability to abstract.

* The evaluation is quite rigorous and showing performance on real world data as well as accounting for human behavior in a relevant task is a very nice touch.

* There are a lot of rigorous proofs in the appendix. The authors have clearly thought a lot about the theoretical foundations of this algorithm as well as shown good empirical proof of its use.

**Weaknesses:**

* I think the paper can mainly be improved in clarity. For example, when getting to Figure 3, it's kind of hard to figure out which exact datasets these results are from. In general, most of the text in the work is dedicated to describing the algorithm and results. I think adding some more information on what datasets are being used would be useful. For example, line 310: " BabyLM language dataset, which contain text snippets from a collection of data domains" what data domains are there? It was hard for me to understand what kind of data this was.

* For Figure 5, I think the authors should plot the difference between the conditions rather than the conditions themselves. Currently, it's hard to understand why HVM shines here more than the LLMs. As I understand, the individual likelihoods don't matter as much as showing a significant decrease from control -> variable conditions. You can put the individual likelihood plots in the supplement. The plot as it is, in my opinion, undersells and obscures the result.

* (minor weakness) I get the feeling that this algorithm isn't quite as scalable as other powerful sequence learning technologies we have today such as LLMs. This is not to say the algorithm is not useful because of that, because of course you do get more interesting structure and interpretability out of it (and it's also a better model of how humans do sequence learning). But I think this is at least worth mentioning in the discussion. If this model does actually have almost as good or better scalability than tools we have today such as SSMs or transformers, I think that would be a huge bonus and definitely needs mentioning.

* (minor weakness) I think a potential missed opportunity for this algorithm is interpreting the discovered variables on specific datasets that we know has rich hierarchical structure. For example, can you use this algorithm on musical notes to recover leitmotifs? But there is enough work here that this can count as future work.

**Questions:**

* What is the relationship between this model and other Bayesian Wake-Sleep class of algorithms, like the older Helmholtz machine or the newer generation such as DreamCoder? There are definitely similarities in having a generative and recognition model, but I didn't see a discussion on this in the paper? I think this would be quite relevant to put in the related works section.

---

> ### Author Response · Authors · 2024-11-23
>
> We thank Reviewer dGcC for their detailed and constructive review. We appreciate the recognition of our algorithm as “really innovative” and the evaluation as “rigorous” with insightful touches on human behavior. We address the comments and questions by the reviewer below.
>
> > I think the paper can mainly be improved in clarity. I think adding some more information on what datasets are being used would be useful. For example, line 310: " BabyLM language dataset, which contain text snippets from a collection of data domains" what data domains are there? It was hard for me to understand what kind of data this was.
>
> We agree that providing more explicit descriptions of the data domains will enhance clarity. Specifically, we have revised the text to explain that the BabyLM dataset includes text snippets from domains such as conversational child-directed speech, narrative stories, and other linguistically rich environments (l 298).
>
> > For Figure 5, I think the authors should plot the difference between the conditions rather than the conditions themselves.
>
> We appreciate the suggestion to reformat Figure 5 for better interpretability. We have updated the plot to show the difference between the control and variable conditions, which more clearly highlights where HVM outperforms the LLMs. We have moved the individual likelihood plots to the supplement.
>
> > I get the feeling that this algorithm isn't quite as scalable as other powerful sequence learning technologies we have today such as LLMs. This is not to say the algorithm is not useful because of that, because of course you do get more interesting structure and interpretability out of it (and it's also a better model of how humans do sequence learning). But I think this is at least worth mentioning in the discussion.
>
> Scalability is indeed a key area for future work. While our current work focuses on cognitive plausibility and interpretability rather than efficiency at large scales, we recognize that scaling up would be a very appealing prospect. We have added a section in the discussion addressing the scalability limitations and potential future directions. Specifically, we propose that HVM could be used as a preprocessing step to extract hierarchical tokens for sequence prediction models, such as transformers, thereby serving as a bridge between interpretability and scalable sequence modeling. We have included additional information about scaling up in the discussion section (l 538).
>
> > (minor weakness) I think a potential missed opportunity for this algorithm is interpreting the discovered variables on specific datasets that we know have rich hierarchical structure. For example, can you use this algorithm on musical notes to recover leitmotifs? But there is enough work here that this can count as future work.
>
> We also appreciate the idea of testing HVM on datasets with known hierarchical structures, such as musical sequences or motifs in classical compositions. While this is beyond the scope of our current work, we agree that such applications could yield valuable insights. We have added this as a direction for future work in the manuscript (l 534).

---

> > ### Comment · Reviewer_dGcC · 2024-11-25
> >
> > Thanks for incorporating my feedback. I think the paper is improved and definitely should be accepted.
> >
> > In terms of my score, I would raise it to a 7 if I could, but I'm not able to do that, so I'm keeping it to be a 6.

---

> > > ### Author Response · Authors · 2024-12-02
> > >
> > > We appreciate your acknowledgement of the improvements made to the paper. We appreciate your intention and feel encouraged to continue developing this line of work.

---

> ### Author Response · Authors · 2024-11-23
> **Relation to the Helmholtz Machine and DreamCoder**
>
> > What is the relationship between this model and other Bayesian Wake-Sleep class of algorithms, like the older Helmholtz machine or the newer generation such as DreamCoder?
>
> Thank you for highlighting the connection to Bayesian Wake-Sleep algorithms. We address the comparison between our model with the two other models below:
>
> __Helmholtz Machine__ The HVM certainly has some relationship with both these models - although they are rather at opposite ends of a spectrum. The Helmholtz machine involves hierarchical generative and recognition models, and is certainly one of the inspirations for the sort of coupling between the two that we have also exploited. However, its hierarchy is not focused on sequences, and, for various reasons, its sequential extension (the Helmholtz machine through time) did not work well, partly because it did not include the sort of iterative processing that we employed in the HVM.
>
> __DreamCoder__ Conversely, the DreamCoder is perhaps best seen as a much more sophisticated and complex recognition model mapping an input into a potentially generating program. Dream coder solves domain-specific tasks by generating programs such as list processing and logo graphics. Dream coder starts with a library of primitive functions that operate on such tasks, and synthesis programs from the library to solve these tasks. DreamCoder contains a wake phase and the sleep phase. During the wake phase, the neural recognition model proposes programs to solve the given tasks. During the abstraction sleep stage, it finds frequently used program combinations as reusable components. During the dreaming sleep stage, the dream coder samples programs from the learned library which defines hypothetical tasks to train the neural recognition model. Dreamcoder and other program synthesis approaches differ from HVM in two primary aspects. One is that the program synthesis approach applies to problems in symbolic tasks in which an input and output is given, different from HVM which takes sequence data. Secondly, the program synthesis approach needs to be initiated with a library of primitive functions, which subtly demands expert knowledge and application domains, whereas HVM does not assume any program primitives and are initialized with an empty dictionary. Thus inference/recognition in the HVM is sufficiently simple that it does not need the extensive machinery inherent to DreamCoder.
>
> We have added discussions about these models to the related work section (l 501).

---

### Official Review · Reviewer_NkSJ · 2024-11-04

**Soundness:** 4
**Presentation:** 4
**Contribution:** 4
**Rating:** 6
**Confidence:** 4

**Summary:**

This paper introduces a hierarchical variable learning model (HVM) that learns abstract patterns from sequences by combining chunking and variable discovery mechanisms. HVM builds on previous chunking models by adding the ability to identify abstract variables that represent groups of chunks that appear in similar contexts. The authors demonstrate that HVM achieves better compression and parsing efficiency compared to baseline models, correlates well with human sequence learning behavior, and provides insights into how abstraction enables better generalization. They evaluate HVM on both synthetic data and real-world language datasets, comparing it against traditional compression algorithms and large language models (LLMs).

**Strengths:**

Impressively, this paper presents some novel theoretical contribution, a clear theoretical framework for combining chunking and abstraction in sequence learning, with formal proofs and guarantees.

They also evaluated their model through multiple angles: computational efficiency, correlation with human behavior, comparison with LLMs, a good set of comparisons.

And I enjoyed their connection to cognitive science: The work bridges computational and cognitive approaches, providing insights into human learning mechanisms.

**Weaknesses:**

1. The paper's comparison to LLMs is relatively narrow and focuses primarily on a specific sequence recall task and limited to short sequences, and therefore seems slightly contrived situation. The paper would benefit from explorations of slightly more complex abstraction tasks to study the general applicability of their method.

2. The comparisons in the paper are quite limited and don't adequately address the rich literature on sequence compression and pattern detection. A single example I have in mind of a similar cogntiively-inspired latent variable learning model is CSCGs (https://pmc.ncbi.nlm.nih.gov/articles/PMC8062558/), but there are more.

**Questions:**

Please answer the two points under weaknesses above.

---

> ### Author Response · Authors · 2024-11-23
>
> We thank Reviewer NkSJ for their thoughtful and constructive feedback. We are glad that our contributions were appreciated, especially the novel theoretical framework and the connection to cognitive science.
>
> > The paper's comparison to LLMs is relatively narrow and focuses primarily on a specific sequence recall task and limited to short sequences, and therefore seems slightly contrived situation. The paper would benefit from explorations of slightly more complex abstraction tasks to study the general applicability of their method.
>
> We appreciate your comment. Our initial aim was to ground HVM's performance in well-understood human sequence learning paradigms. We agree that evaluating HVM on more complex abstraction tasks would strengthen our claims about generalizability and the benefits of hierarchical abstraction. As we do not have data from humans doing a more complex task, in this comparison, we can only stop at the scope of the human experiment. However, the sequences are on a cognitive scale not short, as the length goes beyond working memory span and are challenging to remember. To address this, we propose adding a comparison where HVM and LLMs are evaluated purely on artificial datasets with higher levels of abstraction. Although human performance data would not be available for such tasks, the comparison could highlight differences in learning and transfer complexity between HVM and LLMs. We have added a discussion in the paper and consider including a new table summarizing these comparisons.We could compare HVM with LLM on sequences with more complex hierarchies, in this case the human data might be missing, and the comparison would be purely between the cognitive model and LLMs. Maybe we can have a table about this comparison on the learning and transfer complexity.
>
> > Limited Discussion of Related Work: The comparisons in the paper are quite limited and don't adequately address the rich literature on sequence compression and pattern detection. A single example I have in mind of a similar cogntiively-inspired latent variable learning model is CSCGs (https://pmc.ncbi.nlm.nih.gov/articles/PMC8062558/), but there are more.
>
> We appreciate your comment. We agree that broadening our literature review will improve the paper’s depth.
> George et al. proposed the clone-structured cognitive map as a neural model that replicates observation for different contexts in a cloned higher-order markov chain, and show that CSCG exerts neuronal properties in the hippocampus. This model addresses sequence learning on an algorithmic level and has a neural implementation focus, whereas our model addresses questions on the computational level. We expect future work to bring this computational model of abstract chunk learning to an implementational level.
> We have addressed cognitive models like CSCGs (as mentioned) in the related work section and discuss how our approach differentiates from these models (l 507).

---

> > ### Comment · Reviewer_NkSJ · 2024-11-26
> >
> > I am satisfied with the Author response and maintain my recommendation of acceptance.

---

> > > ### Author Response · Authors · 2024-12-02
> > >
> > > Thank you for your constructive suggestions, support and encouragement. We are grateful for your help to improve this work.

---

### Official Review · Reviewer_RUZK · 2024-11-05

**Soundness:** 3
**Presentation:** 3
**Contribution:** 2
**Rating:** 6
**Confidence:** 3

**Summary:**

**Update after rebuttal:** Thanks to the very detailed response by the authors, my questions and potential misunderstandings have been clarified and I have raised my score from a 5 to a 6. I still think the results on a data source specifically designed to highlight the advantages of the model must be taken with a small grain of salt (recognising that there are other good arguments for designing the data source), and the comparison against LZ78 is okay, but it should not be too surprising that LZ78 is not an ideal compressor for the synthetic data source.

---

The paper tackles the problem of discovering useful abstract representations in sequence prediction, that allow for better compression and transfer (generalization) with the aim of “[capturing] the learning and transfer of abstract representation in human cognition”. To this end, the paper extends a previously proposed model for probabilistic hierarchical chunking, HCM, with the ability to include learnable abstract variables into the chunk dictionary. Learning the model and performing inference with it is somewhat involved (as is generally the case for non-parametric probabilistic models), but explained well in the paper. On synthetic data that is designed to embody the assumptions underlying the model, the model performs very well and outperforms HCM and Lempel-Ziv (LZ78) as a baseline. The model also performs favorably on four natural language datasets (cut into sequences of 1000 characters), and correlates better with human recall times in a color memorization task. The paper finds that in-context learning with LLMs (GPT2 and Llama-2) behaves qualitatively differently on a similar task. Finally, the paper draws a connection between learning abstractions and lossy compression.

**Strengths:**

* The model is an interesting, sensible, and original improvement over HCM.
* Empirical results show that the model learns well and works well on synthetic data and some natural language datasets.
* The work is very well put into wider perspective in the introduction.

**Weaknesses:**

* Many of the main experiments were conducted on data from a generative model that fits exactly the modeling assumptions for HVM (or tasks inspired by these assumptions). As is often the case when a paper proposes a novel model and a novel data set / generator, the fact that the model works well on data that was specifically designed for the model to work well is not a very strong argument. Luckily the paper also shows good results on “natural” data, and qualitatively matches some aspects of human behavior (recall times) that are not trivially predicted from the synthetic data.
* Hierarchical Dirichlet processes and related models (perhaps most famously many variants of topic models) have been used and discussed exhaustively in the ML, linguistics, and cognitive science literature. But an in-depth discussion and technical comparison of HVM against these is currently missing.
* From a cognitive perspective, a severe limitation is that learning in HVM strongly depends on initial learned representations and the order in which learning experiences are represented. While some human learning mechanisms have these traits, I am not sure whether they apply to the learning of abstract concepts in natural language to the extent that the model would predict.


**Verdict:**
The model as an improvement over HCM is sensible, sound, and results clearly show the benefits. Additionally, the cognitive plausibility (at least on an abstract computational level) of the model is decent (at least to me as a non-linguist), and supported by the experiment with human participants. The paper is well written and presents a nice set of experiments. On the other hand, I think many (but not all!) of the experiments must be taken with a grain of salt, since the data has been either synthetically generated to match the model, or has been designed with the same qualities in mind (like in the color memorization task). The comparison against in-context learning in previous-generation LLMs on the same task (translated into text) is ok, but I am not sure whether there is a big take-away other than saying LLMs in-context learning on this task is different from the model and different from human learning on this task. It is unclear whether LLMs should even be designed to mimic HVM in context (the discussion seems to mildly hint at this by claiming normativity of the model). The work is very interesting to a comp. neurosci. audience and a comp. linguistics audience, but its impact in the ML and AI community is likely to be quite limited (nonetheless, a part of the ICLR audience has a background in the aforementioned fields). Some technical discussion around hierarchical Dirichlet processes (or related models) and topic models is missing. Finally, some of the writing and some claims are perhaps a bit overstated (see concrete points under ‘Questions’). In its current state, for an ML conference, I think the relevance and significance of the current work is fairly limited. I think the paper would benefit from a major revision and could be significantly strengthened to be more impactful. I am therefore currently leaning towards rejection (at a top-tier ML conference).

**Questions:**

**Questions and improvements:**

1. The work and claims could be strengthened by evaluating on more datasets that focus on abstraction, but have not been generated by the authors. This is only relevant for a major revision.
2. Topic models and various forms of hierarchical latent variable models have been used and discussed extensively in linguistics, machine learning, and cognitive science. How does the HVM relate to commonly used topic models (LDA, and more modern ones)? Ideally this is discussed on a technical level in detail, but at the least it needs to be included with more detail in the related work discussion.
3. How does the generative model relate to the Hierarchical Dirichlet Process (Teh et al. 2006)?
4. Why was LZ78 chosen as a baseline? It is a lossless general purpose compressor, and has not been designed specifically for natural language data or data with hierarchical structure. Personally I think a much more interesting comparison would be against Context-Tree Weighting (Willems et al. 1995), or maybe the forget-me-not process (Milan et al. 2016), though the latter is perhaps a bit of an overkill (and quite involved to implement).
5. Table 1: “we took random snippets of 1000 characters and calculated evaluation metrics”, that seems like fairly short sequences. Why 1000? Is there a scalability issue with longer sequences? How sensitive are the results, particularly the comparison against LZ78 / CTW to the sequence length?
6. The paper mentions lossy compression multiple times, but as far as I understand all evaluation metrics are lossless in the end (more “lossy” models simply require longer coding lengths / have less coding efficiency)? I am struggling to follow section 4.5 (despite having spent half of my PhD on rate distortion theory). For sure LZ78 is not a lossy compressor - it is lossless. Typically, the distinctive feature of lossy compression is that not all prediction/reconstruction errors matter equally, i.e. the distortion function is typically not the log loss (lossy compression requires a quantitative definition of which information is relevant and to which degree, relative to some task/goal; this is what the distortion function does; the log loss treats all information equally). What is the distortion function in the paper? If one is willing to go lossy, there is a famous trade-off between fidelity and “complexity” (really, the information rate): the rate-distortion curve. I have a hard time relating Fig. 6b to a rate-distortion curve - the “Representation Complexity” seems to be more related to the rate than the distortion, but the figure legend says exactly the opposite. And how is “Abstraction Iterations” (Fig. 6c) related to the abstraction level and representation complexity and thus ultimately the distortion (which also applies to L467-477)? I do agree that lossy compression can be used to formalize a particular kind of abstraction, but it seems to me that what is happening in the paper is more similar to a minimum-description-length argument for *lossless compression* (the more complex the model, i.e. the deeper the tree of abstractions, the better it can compress a sequence, but the price to pay for it is by having a more complex model). The mistake may be fully on my side, but please clarify.
7. Some of the discussion / conclusion is a bit strong. I would not reject the paper based on this, but I have listed concrete issues in the minor comments.
8. Why was the generative model introduced? Were there no suitable generators or datasets in the literature (that are more widely used)? Which shortcomings of previously used data (generators) does the current paper tackle? (I am leaning towards listing this as a minor point, but I also think that any paper that introduces a new data set or data generation procedure should justify it over using what’s been published and used previously by others).

**Minor comments:**
1. L 164 - line break within inline equation.
2. Discussion in L279-284 leaves out that HCM achieves better coding efficiency than HVM if I understand correctly.
3. L 315: “LZ78 performs well in terms of compression efficiency, which is expected given its design purpose”. I don’t fully agree, LZ78 is a general purpose lossless compressor, it has not been specifically developed to compress natural language.
4. L 499: “our work provides a reductionist cognitive model that explicitly specifies the minimal components needed for a model to learn interpretable abstract structure from sequences.” - what makes the model particularly “cognitive”? I also mildly disagree that the model “specifies the minimal components”, rather, it is one solution with few components, but it is unclear that this is the minimum needed (and also minimal in which sense?).
5. L 503: “Our generative mechanism offers a probabilistic, hierarchical sequence generation model relying on chunk-based recursive generation and inventory growth rather than formal grammar rules.” - is this an advantage; does this address some shortcoming in the literature?
6. L 520: “Previously, grammar learning, chunk learning, and statistical/associative learning were studied in isolation as distinct aspects of sequence learning.” - it should be pointed out that this sentence refers to the cognitive science(?) literature (in other fields, like algorithmic information theory, which deals primarily with sequential learning, this distinction does not play a big role).
7. L 523: “Our work suggests a normative origin of concrete and abstract chunk learning” - I think the normative claim is a bit overstated in light of the results and no discussion that rules out all other possibilities.

---

> ### Author Response · Authors · 2024-11-23
>
> We thank Reviewer RUZK for their detailed feedback and constructive suggestions. We are glad the reviewer found the model an "interesting, sensible, and original improvement over HCM" and appreciated the work ‘s connection to neuroscience, cognitive science and linguistic audience. Below, we address the reviewer’s comments, including weaknesses, questions, and minor suggestions.
>
> > Many of the main experiments were conducted on data from a generative model that fits exactly the modeling assumptions for HVM (or tasks inspired by these assumptions)
>
> We acknowledge the reviewer’s concern that some experiments rely on a generative model designed to align with HVM’s assumptions. The generative model was not created to give HVM an unfair advantage but rather to simulate hierarchical structures that are characteristic of certain real-world sequences. This approach mirrors common practices in sequence learning research (e.g., Teh et al., 2006; Milan et al., 2016). Moreover, the generative model itself has merit for studying hierarchical data.  As noted, we also evaluated HVM on natural language data to demonstrate its generalization capabilities beyond synthetic datasets. We have clarified this justification in the revised manuscript (l 220, l 497) and emphasized the importance of showing HVM's robustness on natural datasets as critical validation.
>
> > Hierarchical Dirichlet processes and related models (perhaps most famously many variants of topic models) have been used and discussed exhaustively in the ML, linguistics, and cognitive science literature. But an in-depth discussion and technical comparison of HVM against these is currently missing.
>
> We appreciate the reviewer’s suggestion to compare HVM with other important and influential models. HVM shares an hierarchical nature with a variety of models; however, it differs from topics models, such as Latent Dirichlet Allocation (LDA) by being aimed specifically at sequences rather than treating documents as ‘bags of words’ that can be clustered in a high-dimensional space. It also differs from HDP, which captures clustering over sequences but lacks the library of reusable chunks and variables central to HVM. We have expanded the related work section to include a more detailed discussion of these distinctions (l 503).
>
> > From a cognitive perspective, a severe limitation is that learning in HVM strongly depends on initial learned representations and the order in which learning experiences are represented.
>
> We recognize that HVM's reliance on initial learned representations and order of learning experiences introduces an order dependency. This characteristic, while potentially limiting, is consistent with findings in cognitive science. Human learning mechanisms can be highly order-sensitive and curriculum dependent: the sequence in which information is presented influences learning and memory. In language learning, children acquire more complex compounds as they develop (Newport 1977): stages of child language acquisition follow the geometry of the syntactic tree, with early stages of acquisition corresponding to learning lower parts of the adult syntactic tree, which keeps growing with the development of the child (Friedman et al., 2021, Snow et al.,  1972, and Fernald et al., 1989). This sort of path dependency afflicts any online learning system that lacks an arbitrarily large memory buffer. It is therefore also important for conventional machine learning systems. We have expanded on this characteristic in the discussion, providing references to the cognitive literature, commenting on the statistical limitation, but resource advantage (l520).
>
> Reference:
> 1. Catherine E. Snow. 1972. Mothers’ speech to children learning language. Child Development, 43(2):549– 565.
> 2. Anne Fernald, Traute Taeschner, Judy Dunn, Mechthild Papousek, Bénédicte de Boysson-Bardies, and Ikuko Fukui. 1989. A cross-language study of prosodic modifications in mothers’ and fathers’ speech to preverbal infants. Journal of Child Language.
> 3. Elissa. Newport, Henry Gleitman, and Lila Gleitman. 1977. Mother, id rather do it myself: Some effects and non-effects of maternal speech style. In Catherine E. Snow and Charles A. Ferguson, editors, Talking to Children, pages 109–149. Cambridge University Press.
> 4. Friedmann, N., Belletti, A. & Rizzi, L., (2021) “Growing trees: The acquisition of the left periphery”, Glossa: a journal of general linguistics 6(1): 131.
> 5. Yee Whye Teh. A hierarchical Bayesian language model based on Pitman-Yor processes. 2006
> 6. Kieran Milan, Joel Veness, James Kirkpatrick, Michael Bowling, Anna Koop, and Demis Hassabis.
> The Forget-me-not Process.

---

> > ### Comment · Reviewer_RUZK · 2024-11-24
> > **Thank you for the very detailed response**
> >
> > Thank you for the many clarifications and small additions to the paper, as well as the additional experiments (which are in support of the novel method). I have raised my score and am now leaning towards acceptance.
> >
> > I want to thank the authors for their comments regarding chunking, and the discussion of the other models that I pointed out (topic models, HDP, CTW, forget-me-not). Similarly, the arguments for designing the synthetic data source in the paper are sound (though I still recommend follow-up work to focus on data sources that are not controlled by the authors, if the model is supposed to serve as an explanatory model). The more detailed comments w.r.t. why a comparison against in-context learning in LLMs is particularly relevant make sense. Thank you also for the clarifications regarding rate-distortion - I think it's fine since this is only a minor part in the paper, but I am still struggling a little bit with the notion of lossiness: more abstraction means more uncertainty, agreed, which in the case of *lossless* compression means longer coding length (worse log likelihood) - as in two-part MDL for instance, where typically a more complex model compresses better, but also implies a higher cost due to the model complexity. In lossy compression more uncertainty may or may not incur higher distortion, depending on the distortion measure (one could use log loss as the distortion measure, but then I'm not sure whether the connection to rate distortion adds a lot of insight). Either way, I don't think anything in the paper is factually wrong, so no need to discuss this further.
> >
> > I have no further questions or open issues (that could be addressed within the rebuttal period; of course more empirical verification on other data sources would further strengthen the work).

---

> > > ### Author Response · Authors · 2024-12-02
> > >
> > > Thank you for your detailed and constructive feedback. We agree with your recommendation for the future work to apply additional sources to further demonstrate the model’s explanatory power in real-world settings.
> > >
> > > Regarding the discussion on rate-distortion and the notion of lossiness, while our current work points out the relation between abstraction, the uncertainty it introduces, and its influence on generalization, we acknowledge that future work could benefit from a deeper exploration of distortion measures and their implications. We aim to theoretically investigate this area in subsequent studies.

---

> ### Author Response · Authors · 2024-11-23
> **Limited Relevance on the ML and AI Community**
>
> > I am not sure whether there is a big take-away other than saying LLMs in-context learning on this task is different from the model and different from human learning on this task. ... The work is very interesting to a comp. neurosci. audience and a comp. linguistics audience, but its impact in the ML and AI community is likely to be quite limited (nonetheless, a part of the ICLR audience has a background in the aforementioned fields).
>
> Thank you for raising the question about the implications of different behavior of LLMs, humans, and HVM on the cognitive task. This section is included in the paper because LLMs have been reported as having emergent abilities such as solving mathematical problems and reasoning tasks (Webb 2023, Kojima 2022). This has raised the (popular, but controversial) idea that given sufficient training LLMs may exceed human intelligence . However, controversy has accompanied the findings that LLMs do not learn to generalize on symbolic tasks, and particularly those that involve abstract representation (Trott 2022, Ransom 2022) and compositional operations (Dentella 2024).
>
> It therefore becomes especially important to use a task that unearths the aspects of LLM that differ from human cognition, and find out whether there are features inside a more conventional cognitive model that might provide the missing component. Our observations about the match between human data and the HVM suggest make it a candidate. Along similar lines, the experiment can be adapted to evaluate other types of model for their ability to abstract. Therefore, we expect this task to appeal to a broad audience beyond cognitive scientists and linguists.
> We have also updated our experimental comparison to include more recent large language models.
>
> To clarify this point, we have also changed the LLM section to substantiate the importance of comparing LLM with humans and cognitive models (l 374). We have also highlighted HVM’s potential as a preprocessing tool for tokenization in neural sequence models, enabling them to leverage hierarchical abstractions (l 522).
>
> Reference:
>
> 1. Kojima, T., Gu, S.S., Reid, M., Matsuo, Y., Iwasawa, Y.: Large language models are zero-shot reasoners. NeurIPS 2022
> 2. S. Trott, C. Jones, T. Chang, J. Michaelov, B. Bergen, Do large language models know what humans know? arXiv [Preprint] (2022).
> 3. Han, S. J., Ransom, K. J., Perfors, A., & Kemp, C. (2022). Human-like property induction is a challenge for large language models. In Proceedings of the annual meeting of the cognitive science society (Vol. 44, No. 44).
> 4. Dentella, V., Günther, F., Murphy, E. et al. Testing AI on language comprehension tasks reveals insensitivity to underlying meaning. Sci Rep 14, 28083 (2024).
>
>
> > The work and claims could be strengthened by evaluating on more datasets that focus on abstraction, but have not been generated by the authors. This is only relevant for a major revision.
>
> Thank you for the comment. We are aware of datasets such as BIG-bench, that evaluates abstraction by computing similarities between multiple-choice words based on text queries (Srivastava et al. 2023), or the ARC challenge which is designed with symbolic reasoning in a graphical setting. Both these benchmarks operate in domains that are more complicated than the simple sequence setting that we study. Furthermore, our algorithm is too simple to be plugged directly into abstraction tasks that are based on language description or symbolic reasoning. We have taken this suggestion as a reference for future work to apply such models on benchmarks.
>
> > Topic models and various forms of hierarchical latent variable models have been used and discussed extensively in linguistics, machine learning, and cognitive science. How does the HVM relate to commonly used topic models (LDA, and more modern ones)? Ideally this is discussed on a technical level in detail, but at the least it needs to be included with more detail in the related work discussion.
>
> Please see our brief discussion above comparing the sequence learning models proposed with topic models (l 511). We have expanded the related work section and the discussion to make these points explicitly, within the confines of the length limit.
>
> > How does the generative model relate to the Hierarchical Dirichlet Process (Teh et al. 2006)?
>
> The HDP is a very attractive model of sequences - which is even used by some to characterize everything from sequence learning in humans to the structure of the songs of Bengalese finches. However, HVM differs from hierarchical Dirichlet Process (HDP) models in the nature of its library (which is more flexible than in the HDP); and it also differs from fragment grammar methods by having the notion of variables that are ‘interior’ to chunks. Thus, HVM is endowed with the sort of flexibility that is critical for characterizing sequences with rich hierarchical structure, including natural languages.

---

> > ### Author Response · Authors · 2024-11-23
> >
> > > Why was LZ78 chosen as a baseline? It is a lossless general purpose compressor, and has not been designed specifically for natural language data or data with hierarchical structure. Personally I think a much more interesting comparison would be against Context-Tree Weighting (Willems et al. 1995), or maybe the forget-me-not process (Milan et al. 2016), though the latter is perhaps a bit of an overkill (and quite involved to implement).
> >
> > We conducted model comparison on dictionary-based sequence learning models. As Context-tree weighting, forget-me-not process do not learn a dictionary from sequences, we included a discussion about how these models relate to our work in the paper.
> >
> > > Table 1: Why 1000? Is there a scalability issue with longer sequences? How sensitive are the results, particularly the comparison against LZ78 / CTW to the sequence length?
> > 	Thank you for your comment, we conducted additional model comparison experiments using sequences of different lengths.
> >
> > Seql = 2000
> >
> > | Model         | Parsing Length | Negative Log Likelihood |
> > |---------------|----------------|--------------------------|
> > | HCM           | 5.49           | 1652                    |
> > | HVM           | 6.84           | 1474                    |
> > | Ground Truth  | 6.38           | 1132                    |
> > | LZ78          | 5.27           | 3124                    |
> >
> > ---
> > Seql = 3000
> >
> > | Model         | Parsing Length | Negative Log Likelihood |
> > |---------------|----------------|--------------------------|
> > | HCM           | 4.11           | 3093                    |
> > | HVM           | 5.62           | 2739                    |
> > | Ground Truth  | 6.38           | 1819                    |
> > | LZ78          | 5.75           | 4574                    |
> >
> > ---
> > Seql = 4000
> >
> > | Model         | Parsing Length | Negative Log Likelihood |
> > |---------------|----------------|--------------------------|
> > | HCM           | 3.53           | 4600                    |
> > | HVM           | 7.98           | 2961                    |
> > | Ground Truth  | 6.38           | 2425                    |
> > | LZ78          | 6.39           | 5813                    |
> >
> >
> > > ‘the log loss treats all information equally, lossy compression requires a quantitative definition of which information is relevant and to which degree, relative to some task/goal; this is what the distortion function does’ ‘And how is “Abstraction Iterations” (Fig. 6c) related to the abstraction level and representation complexity and thus ultimately the distortion (which also applies to L467-477)?’
> >
> > We thank the reviewer for raising this point. With more iterations of abstraction, more variables will be created and defined on the basis of existing chunks/variables, the number of variables learned grows with the number of iterations, as suggested by Fig. 6a. This implies a trade-off between fidelity and complexity as the iteration increases.
> >
> >  The deeper the tree of abstractions, the better it can compress a sequence . That is, higher order variables provide explanations for longer parts of the sequence. This is evident in our use of parsing length as the rate function: the number of chunks inside the dictionary that are needed to describe a particular sequence. However, the price to pay is the greater uncertainty inherent in the variables embedded inside the chunks.  The chunks learned at later iteration steps contain higher order variables, and carry inherent variability when just the chunks are used to describe the sequence.  The use of more variables will introduce more distortion, because they leave uncertainty as to which particular units are in the sequence.
> > We refer to the resources needed to encode the variables within the chunks as the representational complexity of a sequence  $RC(\mathbb{V}) = \sum_{v \in \mathbb{V}} \sum_{ u\in \mathbb{E}(v)} -\log P(u|v)$. Higher representational complexity implies a more refined and nested definition of variables. In the extreme case without abstraction, the representational complexity is 0, which is the lossless compression (corresponding to when the model only learns chunks).
> >
> > Since the model learns abstraction, one iteration at a time, the patterns that the model uses to parse the sequence will be bigger after the 20th than the 1st iteration. However, the chunks used will also have higher representational complexity.
> > The relation between the RD curve and generalization is shown in Figure 6c. Indeed, as the reviewer said, not all distortions are equal. HVM involves a lossy compression process. Higher level abstraction will help the model to parse novel sequences with lower negative log likelihood.
> >
> > We have revised the manuscript to convey a clearer connection between the distortion introduced by learning layers of abstraction, and the lower rate the abstraction process introduced.

---

> > > ### Author Response · Authors · 2024-11-23
> > >
> > > > Why was the generative model introduced? Were there no suitable generators or datasets in the literature (that are more widely used)? Which shortcomings of previously used data (generators) does the current paper tackle?
> > >
> > > We introduced our own generative model because, indeed, we could not find suitable generators or datasets in the literature. We are suggesting that the world contains nested hierarchical structure, with chunks as a central generative force. Although chunks were once a popular focus of heuristic recognition models, and they have some resonance in modern generative models such as fragment grammars, they have not been a substantial focus of study in recent times. The other statistical models have concentrated on other facets, with the hierarchical dirichlet process focusing on the power-law aspect of natural languages (Teh et al. 2006); the forget me not process (Milan, 2016) modeling data sequences with abruptly changing statistics, and CTW modeling sequences as a stochastic chain. We think that our generative model merits further investigation. We have edited the discussion section (l 496) and the generative model section (l 88) to make this clearer.
> > >
> > > > ‘Discussion in L279-284 leaves out that HCM achieves better coding efficiency than HVM if I understand correctly.’
> > >
> > > Yes, this is because HVM learns a bigger dictionary.
> > >
> > > > L 499: “our work provides a reductionist cognitive model that explicitly specifies the minimal components needed for a model to learn interpretable abstract structure from sequences.” - what makes the model particularly “cognitive”?
> > >
> > > The model is termed cognitive as the design is inspired from human chunking. The minimal components referring to the model parts are based on what we know about what humans can learn. The model records first order transition probabilities because humans are sensitive to first order transitions in sequences (Saffran 1996, 1999, Aslin and Newport 2012). On top of that, the model learns chunks because humans learn chunks (Perruchet and Vinter 1998). Finally, the model proposes variables as symbols because these symbols that represent categories have been observed in other cognitive studies (Marcus 1999). Additionally, we show that the model’s sequence learning behavior resembles human recall time. The above aspect about the design architecture and the learning effect of the model makes it cognitive.
> > >
> > > > L 503: “Our generative mechanism offers a probabilistic, hierarchical sequence generation model relying on chunk-based recursive generation and inventory growth rather than formal grammar rules.” - is this an advantage; does this address some shortcoming in the literature?
> > >
> > > Our generative mechanism compliments models that rely on predefined rigid grammatical structures, which often struggle to capture the adaptive and evolving nature of real-world sequential data. By focusing on probabilistic chunking and hierarchical organization, our model better mirrors the flexible learning and processing mechanisms observed in human cognition and offers improved generalizability for tasks involving the extraction and understanding of complex, unknown patterns.
> > >
> > > > L 520: “Previously, grammar learning, chunk learning, and statistical/associative learning were studied in isolation as distinct aspects of sequence learning.” - it should be pointed out that this sentence refers to the cognitive science(?) literature.
> > >
> > > Thank you for your suggestion. Indeed, they refer to the cognitive literature. We have specified this in the updated text (l is 521).
> > >
> > > > L 523: “Our work suggests a normative origin of concrete and abstract chunk learning” - I think the normative claim is a bit overstated in light of the results and no discussion that rules out all other possibilities.
> > >
> > >  Thank you for your suggestion, we have toned down the statement.
> > >
> > > Reference
> > >
> > > 1. Saffran, J. R., Aslin, R. N., & Newport, E. L. (1996). Statistical learning by 8-month-old infants. Science
> > > 2. Saffran, J. R. (1999). Statistical language learning: Mechanisms and constraints. Current Directions in Psychological Science
> > > 3. Aslin, R. N., & Newport, E. L. (2012). Statistical learning: From acquiring specific items to forming general rules. Current Directions in Psychological Science
> > > 4. Perruchet, P., & Vinter, A. (1998). Learning and development: The implicit knowledge assumption reconsidered. Cognitive Development
> > > 5. Marcus, G. F. (1999). Rule learning by seven-month-old infants. Science
> > > 6. Teh, Y. W., Jordan, M. I., Beal, M. J., & Blei, D. M. (2006). Hierarchical Dirichlet processes. Journal of the American Statistical Association
> > > 7. Milan, S. (2016). The Forget-me-not process NIPS 2016
> > > 9. Willems, F. M. J., Shtarkov, Y. M., & Tjalkens, T. J. (1995). The context tree weighting method: Basic properties. IEEE Transactions on Information Theory

---

### Author Response · Authors · 2024-11-23
**General Response**

We would like to thank all reviewers for their constructive and detailed feedback. The overall assessment of our work has been positive, with reviewers appreciating the novelty of the Hierarchical Variable Model (HVM), its cognitive relevance, and the rigorous evaluations conducted.
Some highlights from the reviews include:
* Reviewer NkSJ commended the paper for its “novel theoretical framework” and appreciated the “clear connection to cognitive science”.
* Reviewer dGcC described the algorithm as “really innovative” .
* Reviewer RUZK appreciated the originality of the model, calling it an “interesting, sensible, and original improvement over HCM”, stating that the paper is “relevant to audiences in neuroscience, cognitive science, and linguistics.”
* Reviewer z6tL noted that “the paper is organized and well-written” with “significant contributions to the field of abstraction in AI”.

In response to the reviewers’ constructive comments, we have made the following modifications to our manuscript:

* We have further highlighted the importance of studying the difference between LLM and human cognition in the LLM section of the paper.
* We have included discussions comparing HVM to models such as hierarchical Dirichlet processes, context-tree weighting, the forget-me-not process, CSCGs, Helmholtz machines, and DreamCoder.
* We have included additional experiments, such as comparisons with more recent LLMs (e.g., Llama 3), and sensitivity analyses to assess the robustness of HVM’s performance across varying parameter settings.

We believe these revisions have strengthened our manuscript during the review process. We again thank the reviewers for their valuable input, which has substantially improved the quality and scope of our work.

---

### Meta-Review · Area_Chair_vjTB · 2024-12-20

**Metareview:**

This paper introduces the Hierarchical Variable Learning Model (HVM), a novel extension of the Hierarchical Chunking Model (HCM) that learns abstract representations by grouping sequence chunks into higher-level categories based on contextual similarity. This enables HVM to achieve improved compression, parsing, and generalization compared to baselines (learned baseline = HCM, hard-coded baseline for compression = LZ78), and to better correlate with human behavior in sequence memory tasks. The model is evaluated on synthetic data and BabyLM (real language data). The paper also presents a human study in a sequence memory task, and compares people's performance with HCM, HVM and a variety of LLMs. HVM better captures human performance.

This paper is most compellingly read as a contribution to computational cognitive science -- the results are informative about how people might learn chunked, abstract representations, and the paper puts forth a model which better captures this behavior relative to alternatives. Reviewers note that this is both a strength and a weakness. As a strength, HVM is novel, rigorously and thoroughly evaluated, and seemingly a good model of human behavior. As a weakness, there is limited evidence that HVM would be useful in more naturalistic datasets which are more important in AI / ML (although the authors added discussion of this to the paper during the rebuttal period).

Despite the uncertainty raised about the relevance of this work for an AI / ML audience, all reviewers voted to accept this paper after the rebuttal period. I similarly feel that the paper makes a compelling contribution by demonstrating a novel method which can learn abstract, chunked representations, and therefore deserves to be presented at ICLR.

**Additional Comments On Reviewer Discussion:**

As noted above, several reviewers brought up potential issues in applying HVM to naturalistic datasets, which the authors now include as part of a discussion of future work. Similarly, reviewer RUZK rightly pointed out missing related work to hierarchical dirichlet processes (HDPs), which the authors added to the paper.

Reviewer dGcC had several suggestions for improved clarity, which the authors generally incorporated into their revision.

---

### Decision · Program_Chairs · 2025-01-22

Accept (Poster)